# The psychosocial impact of exercise as an intervention for persons living with obesity and female infertility: A rapid scoping review and pilot study

Tiffany Furneaux[1], Jillian Murdoch[1], Nicole Hollohan[1], Catherine M. Barrett[2], Deanna Murphy[1], Alison Farrell[1], Erin McGowan[2], Laurie K. Twells[1], Katie P. Wadden[2]*

**1** Faculty of Medicine, Memorial University of Newfoundland, St. John's, Newfoundland and Labrador, Canada, **2** School of Human Kinetics and Recreation, Memorial University of Newfoundland, St. John's, Newfoundland and Labrador, Canada

* kwadden@mun.ca

## Abstract

Persons living with obesity and experiencing infertility are counselled on healthy behaviours, such as increasing physical activity levels, to improve fertility-related outcomes. However, due to the known psychological burden of receiving an infertility diagnosis, there is an important need to consider the psychosocial impact of administering exercise programming for this population. This study aims to assess the psychosocial impact of exercise-based interventions on individuals with obesity and infertility through a rapid scoping review and a pilot study. The rapid scoping review was conducted in MedLine, Embase, and CINAHL. Studies involving exercise-based lifestyle interventions for people with obesity and infertility were included only if they measured a psychosocial outcome as a primary or secondary measure. Expanding upon evidence from our rapid review, a pre-experimental feasibility study (pre-test, post-test with one group) was conducted. The pilot study implemented a virtual exercise intervention for people with obesity and experiencing infertility. Questionnaires measuring anxiety and depression, quality of life (QoL), social support, fertility-related stress, and hopelessness were administered. Based on the nine articles from our rapid review, there is evidence that lifestyle interventions combining exercise, diet, and psychobehavioral components improve psychosocial outcomes like anxiety, depression, QoL, and self-esteem. In our pilot study, 11 participants (Age: 34±3.7, BMI: 40.3±4.54) engaged in the virtual exercise intervention, and seven completed the post-intervention questionnaires. The results of our pilot study showed improvements in depression, hopelessness, physical QoL, and fertility-related stress scores. However, we observed declines in anxiety, mental QoL, and perceived social support measures. Our mixed findings may be due to the unique context of our pilot study. The study took place during the COVID-19 pandemic, where increased stress,

**Data availability statement:** Data can be accessed: Furneaux, Tiffany Jessica Josephine, 2024, "Replication Data for: The Psychosocial Impact of Exercise as an Intervention for Persons Living With Obesity and Female Infertility: A Rapid Scoping Review and Pilot Study", https://doi.org/10.7910/DVN/M8W8CV, Harvard Dataverse, V1.

**Funding:** This work was supported by The Newfoundland and Labrador (NL) Healthcare Foundation Research Grant (20210286 to TF), Mitacs Accelerate Fellowship (20210629 to TF), and the Memorial University of Newfoundland Rita Love Foundation Grant (20210167 to TF), NL Support Patient-Oriented Research Unit Healthcare Professional Project Grant (20201121). The funders had no role in study design, data collection and analysis, decision to publish, or preparation of the manuscript.

**Competing interests:** The authors have declared that no competing interests exist.

anxiety, and isolation, negatively impacted fertility patients. Additionally, the virtual intervention, required by restrictions, may have reduced social interaction and support, worsening one's mental health.

## Background

Infertility is defined as the inability to conceive after a year of sexual intercourse without contraception [1]. Infertility rates have doubled since the 1980s and currently affect approximately one in six Canadian couples [2]. The experience of infertility can have psychosocial consequences, such as elevated levels of stress and anxiety and the experience of symptoms of depression and feelings of hopelessness [3–6]. Women experiencing infertility are more likely to experience higher levels of anxiety [7], depression [8], and psychological distress [9] compared to women not experiencing infertility. Furthermore, there is evidence to suggest that increased stress could negatively impact unassisted conception rates and fertility treatment outcomes [10–12]. Unfortunately, it appears the relationship between reproductive health and psychosocial factors is bidirectional, meaning these factors often increase and further impact fertility following an infertility diagnosis [7–10]. The severe psychological impact of infertility is emphasised in a study where women with infertility reported anxiety and depression levels similar to those of women with cancer, heart attacks, and HIV [13]. Thus, there is an important need to measure and monitor the psychological well-being of patients being treated for infertility.

Individuals diagnosed with infertility who live with obesity face distinct psychological challenges during reproductive care. Due to the connection between excess adiposity and dysfunction in processes related to fertility, patients with obesity are often counselled by healthcare providers to reduce their body weight [14]. However, there is limited research that examines the psychosocial impacts on persons with obesity and infertility who have been prescribed lifestyle interventions [15]. A recent study explored the experiences of Canadian adults who live with two or more chronic diseases with healthcare providers incorporating obesity treatment guidelines in chronic disease management [16]. The study found that more than half of respondents who received a diagnosis of infertility perceived the obesity management support they received as "not very helpful" or "not at all helpful" [16]. The impact of advising patients to lose weight (often a considerable amount) without adequate support can lead to adverse health outcomes such as medication nonadherence, mistrust of their healthcare provider and avoidance of medical care [17,18]. To improve adoption and adherence to programs focused on behavioural change and minimize the negative consequences, there is a critical need to examine psychosocial outcomes for persons with female infertility and obesity during reproductive care.

Independent of weight loss, physical activity and exercise are critical components of a lifestyle intervention for improved cardiometabolic health in preconception and pregnancy [19]. The primary objective of the present study is to determine the psychosocial effect of exercise as an intervention, or a component of an intervention,

for persons of reproductive age (ages 18–40) living with obesity (BMI> or equal to 30 kg/m^2) and female infertility. Our research question is assessed in two ways. First, through a rapid scoping review describing the methods and psychosocial effects of exercise-based lifestyle interventions in persons with obesity and experiencing infertility. And second, through the results of a pre-experimental study examining self-reported measures of anxiety, depression, hopelessness, infertility-related stress, and perceived social support before and after a 12-week virtual exercise intervention in persons living with obesity and experiencing female infertility. Based on our review of the literature, we expected improvements in all measured psychosocial parameters. However, this pilot feasibility study did not have statistical power for hypothesis testing.

We chose to combine a rapid scoping review with a pilot study as a strategic approach to effectively explore an under-researched area [20]. The rapid review aimed to identify gaps in the literature to inform our research objectives and our pilot study focused on exploring psychosocial measures and identifying challenges in implementing an exercise-based intervention for individuals with obesity and infertility, providing real-world context to theoretical insights. Together, the rapid scoping review and pilot study provide insights into the psychosocial effects of undergoing an exercise-based lifestyle intervention when living with obesity and infertility.

## Methods

We utilized a dual methodological approach by combining a rapid scoping review and a pilot study to strengthen our validity of our findings while addressing unmet needs in the literature. Of note, this study used patient-oriented research methods, informing the design of the pilot study to ensure the study's objectives and methodology were meaningful to our patient population [21].

### Ethics statement

This study received ethics approval from the provincial health research ethics authority (HREA # 2019.214).

### Rapid scoping review

This rapid scoping review was started on June 14th, 2023, with the search being repeated in December 2023 to review any recently published articles. This rapid scoping review aimed to: (1) explore existing studies which evaluated exercise-based interventions for participants with obesity and experiencing infertility, and (2) to understand the psychosocial effects of these interventions on the target population. Search components and terms were discussed amongst the team, with a final search being developed by a health sciences librarian (AF) using Mesh and keywords in Ovid Medline. The search was tested to ensure key articles were discovered. The search was then performed in Ovid Medline, Embase.com, and CINAHL (Ebsco). The search strategy for Medline sets out the search approach in more detail (S1 Text). Results were imported into and duplicates were removed using Covidence software.

Articles were screened by one author (JM). Articles identified through the initial search were first screened based on title and abstract. Articles not fulfilling at least one inclusion criteria were excluded at this stage. Next, articles were screened through full text review. Articles were included in the review based on the following inclusion criteria: (1) participants were experiencing female-factor infertility; (2) participant BMIs were greater than or equal to 30 kg/m^2; (3) participants were seeking fertility services; (4) interventions included an exercise component; and (5) the reported outcomes included psychosocial measures such as anxiety, depression, quality of life, fertility related stress, and hopelessness. Articles not meeting all inclusion criteria, not published in the english, or not available in full text form were excluded. were excluded. Predetermined details from each study were then extracted from each study. JM and KW completed the Cochrane Risk of Bias assessment of each study using need to the following five domains: (1) bias arising from the randomization process; (2) bias due to deviations from intended intervention; (3) bias due to missing outcome data; (4) bias in the measurement of the outcome; (5) bias in the selection of the reported result. Judgement of each study was based

on the following rating system: high, some concerns, low and no information. KW acted as the second reviewer and verified all judgments. This review's risk of bias assessment is available in (S2 Text).

The data extracted from the articles in this review included population data (age, sample size, BMI, and cause of infertility); intervention data; and outcome data (psychosocial outcomes measured and associated results). Data was extracted and charted by one author (JM). The PRISMA Scoping Review Checklist for this review is available as a supplementary file (S1 Checklist).

### Pilot study

**Pre-experimental study with patient-oriented research approach.** A feasibility assessment of this pilot study is reported in our previously published work on the same cohort [22]. This study used a patient-oriented research approach, which involved including people with lived experience (PWLE) as collaborating research partners, in the development of this study's objectives and methodology. The study methodology was informed by four PWLE collaborators who have lived experience of obesity and infertility. Two patient engagement sessions were held to elicit patient-partner input. During the first session, the researchers consulted the PWLE collaborators on the research topic to engage in an exchange of information to connect the PWLE collaborators' experience with the researchers' knowledge. Based on patient partner responses to the researcher's questions, the study protocol was modified to include questionnaires to measure psychosocial variables. During the second patient engagement session, PWLE collaborators reviewed the developed methodology. Researchers documented the PWLE collaborators' feedback and revised the methods based on their recommendations.

During discussion group sessions with PWLE collaborators, several key emotional and psychological experiences were shared, including anxiety, depression, hopelessness, infertility-related stress, and perceived social support. From these discussions, the researchers selected validated questionnaires to measure the constructs identified by the patients. The questionnaire package included six validated tools: (1) Leisure Time Physical Activity Questionnaire, (2) Hospital Anxiety and Depression Scale (HADS), (3) Short Form-12 Version 2 (SF-12v2), (4) Beck Hopelessness Scale (BHS), (5) Fertility Problem Inventory (FPI), and (6) Multidimensional Scale of Perceived Social Support (MSPSS) [22–27]. This paper reports on results from questionnaires 2–6, as the results from questionnaire 1 are reported elsewhere [22].

Scores for depression and anxiety were measured using the Hospital Anxiety and Depression Scale (HADS) [24]. The HADS has been validated for patients and the general public [28]. The HADS questionnaire has seven items each for depression and anxiety. Scoring for each item ranges from zero to three, with three denoting the highest levels of anxiety or depression. Total scores range from 0-21, with higher scores representing more severe levels of depression and anxiety. Total subscale scores for depression and anxiety equal to or above eight represent considerable symptoms of depression and anxiety.

Health-Related QoL was measured using the SF-12v2. The SF-12v2 is a health-related quality-of-life questionnaire consisting of twelve questions that measure eight domains of health that assess physical and mental health. The SF-12v2, a shortened version of the SF-36 is a valid assessment of the same domains with fewer questions to reduce the respondent burden amongst participants [29]. Based on survey questions of eight domains, physical and mental component scores can be calculated with scores ranging from 0-100 and 50 (±10), indicating normative scores for the general population.

Fertility-related stress was measured by using the Fertility Problem Inventory (FPI) [26]. The FPI has demonstrated validity in women experiencing infertility [30]. The 46-item questionnaire uses a Likert scale from 1 to 5 to measure the following five domains: 1) sexual concern, 2) social concern, 3) relationship concern, 4) need for parenthood, and 5) rejection of a childfree lifestyle. Using the FPI domain scores, a measure of global stress can be calculated by summing the individual scores of the five domains. For global fertility-related stress, the scores can range from 46 to 276. Higher scores signify higher levels of fertility-related global stress.

Hopelessness was measured using the Beck Hopelessness Scale (BHS) [22]. The BHS is a 20-item self-reported questionnaire designed to quantify three significant aspects of hopelessness in outpatients: feelings about the future, loss of motivation, and expectations. BHS has high predictive and internal validity [31]. Scores range from 0 to 20. Scores from 0-3 are considered in the normal range, those from 4-8 equate to mild hopelessness, and scores between 9–14 represent moderate hopelessness. Scores above 14 identify severe hopelessness.

The Multidimensional Scale of Perceived Social Support (MSPSS) was developed to obtain subjective measures of perceived social support from three sources: family, friends, and significant others [27]. The MSPSS has been validated for use in various populations [32]. The MSPSS includes 12 items scored on a Likert scale from 1-5 [27]. Scores range from 12-84, the higher the score, the higher the perceived social support. Scores from 12-35 signify low levels, 36–60 medium levels and 60–84 high levels of social support.

**Recruitment.** Participants were recruited primarily through physicians at a local fertility clinic. Potential patients were identified by physicians: (1) as new patients to the NL Fertility Services clinic for their initial assessment or (2) following a review of current patients' medical records by a physician in their circle of care. Social media platforms (local Facebook Support Groups, such as "Faces of Fertility" and "Newfoundland and Labrador Infertility Support") were used as an additional means of recruitment using a recruitment poster. A total of 32 persons with female infertility were referred to the research team to discuss the details of the study. A research team member called and emailed all potential participants to introduce the study. After initial referral, 21 of the contacted persons decided not to participate for one of the following reasons: (1) not interested at this time, (2) do not have enough time to commit during the week, (3) do not want to potentially delay fertility treatment to participate, (4) starting their weight-loss program (Some patients decided not to participate as they had already begun a weight-loss program, independently), or (5) lack of confidence.

Participants were recruited based on the following inclusion and exclusion criteria. The inclusion criteria for the present study were: (1) between the ages of 18 and 45 experiencing infertility (i.e., inability to conceive after twelve months of trying) through either self-report or physician referral, (2) with a BMI > 30 kg/m$^2$, (3) who were not meeting the Canadian Physical Activity Guidelines for physical activity, and (4) who are willing to commit to an online group exercise program three days a week for 12-weeks. Women were excluded from the study if they were < 18 or older than 45 years, had physical impairments limiting their ability to participate, or were unwilling to delay fertility treatment for 16-weeks. The exclusion criteria were selected to ensure clear, interpretable results focused on the effects of an exercise intervention on women with obesity (BMI > 30 kg/m$^2$).

Participants who agreed to participate in the study were emailed a consent form, a Physical Activity Readiness Questionnaire for Everyone (PARQ+), and the Godin-Shephard Leisure-Time Physical Activity Questionnaire to determine eligibility. Upon revision of the PARQ+ responses, participants who indicated having specific comorbidities like asthma or high blood pressure were given a PARmed-X to obtain medical clearance from their healthcare provider. The Godin-Shephard Leisure-Time Physical Activity Questionnaire was used to determine whether participants met the eligibility criteria for physical activity while also being used as a validated tool for data collection mentioned below. After one month of recruitment, eleven participants were successfully enrolled in the study.

**Intervention.** The details of the exercise intervention are outlined in our previous publication, known as the PRO-FIT-CARE study [33]. The exercise intervention was designed for participants to exercise at a moderate-to-vigorous intensity level while completing low-impact, body-weighted exercise [33]. Due to the COVID-19 pandemic, the exercise intervention was in an online group setting environment using Zoom Software. The exercise sessions were delivered live by a registered Kinesiologist three days a week, each lasting 45 minutes and spanning 12 weeks. Live participation was encouraged; however, sessions were recorded for participants unable to attend to complete at a later time. A private Facebook page for participants to engage with one another was also created based on feedback from the patient partner discussion groups. On this Facebook page, members of the research team posted nutritional tips and recipes, motivational quotes, and, most importantly, links to the virtual exercise program.

**Data collection.** Participants completed a questionnaire package pre-intervention and after the 12-week intervention. Questionnaires were sent to participants' emails or printed and delivered to their home addresses. Due to the virtual nature of the program, all questionnaires were completed by the participant and contained self-reported data. The following questionnaires were administered: 1) Godin-Shephard Leisure Time Physical Activity Questionnaire (GSLTPAQ); 2) Hospital Anxiety and Depression Scale (HADS); 3) Short Form Health Survey, version 2 (SF-12v2)); 4) Fertility Problem Inventory (FPI); 5) Beck Hopelessness Scale (BHS); 6) Multidimensional Scale of Perceived Social Support (MSPSS). In addition, information was gathered through a post-intervention semi-structured discussion group. Last, anthropometric measures (height and weight) were self-reported, year of birth, current medications, and comorbidities were collected.

Data was collected using virtual and physical copies of the questionnaires. Individual scores were inputted into a Microsoft Excel spreadsheet, separated by the questionnaire used and variable measured. Due to the small sample size (n = 11), statistical analysis of pilot study data is limited to descriptive statistics. Confidence intervals and P-values for the results are not reliable due to the small sample size of the pilot study [34]. A student license was obtained from Quality-Metric to analyze the SF-12v2 data. PRO CoRE scoring software was used to score SF-12v2 domains and to analyze means and standard deviations of the physical and mental component scores (PCS, MCS, respectively).

**Patient satisfaction questionnaire.** To gather more information about patient's satisfaction with exercise intervention, using Google forms, participants responded weekly to open-end questions regarding the program, instructors, how participants felt during the week and any other general comments. Impactful quotes from these responses underwent thematic analysis by two reviewers (CB, KW) and were categorized into subthemes and themes [35].

## Results

### Rapid scoping review

In the initial search, 838 studies were identified through database searching and were imported to Covidence. After duplicates were removed, the remaining 662 articles were selected for title and abstract screening. Of these articles, 18 were uploaded to full-text screening for assessment of review eligibility. Ultimately, nine articles were reviewed in the rapid scoping review (Fig 1).

The nine articles reviewed discussed a total of four different intervention protocols. Relevant information about study samples, interventions and results are summarised in Table 1. Three of the articles provide only the study intervention protocol information [36–38]. The remaining six articles report on results from the selected studies [15,39–43]. All studies investigated the psychosocial effects of exercise-based interventions for women with obesity and experiencing infertility. The selected studies were based in Australia [39,40], the Netherlands [15,37,38,41,43], Spain [36], and the United-States of America [42].

Of the completed studies reviewed, there was one cohort study [39,40], two multi-centre randomized control trials [15,37,38,41–43]. The interventions of included studies varied in duration and in concept. The intervention duration of included studies varied between 12 and 24 weeks. The intervention concepts also varied. In Galletly et al's 1996 study there was a weekly session with a psychiatry, dietetics, or reproductive medicine specialist to discuss the effects of life-style changes on pregnancy outcomes alongside their exercise intervention [39,40]. The FIT-PLESE intervention included a calorie restriction and anti-obesity medication component alongside their exercise component [42]. Last, the LIFEstyle study intervention included dietary change and behaviour modification components alongside their exercise component [15,37,38,41,43]. Further, the exercise component of the respective interventions also differed. For example, Galletly et al's study included a weekly, hour-long, group exercise component while the FIT-PLESE and LIFEstyle studies had exercise interventions more focused on obtaining weekly or daily step counts.

The outcomes measured in these completed studies varied as well. Galletly et al's study measured anxiety, depression and self esteem while the FIT-PLESE and LIFEstyle studies focused more on quality of life. Galletly et al found statistically

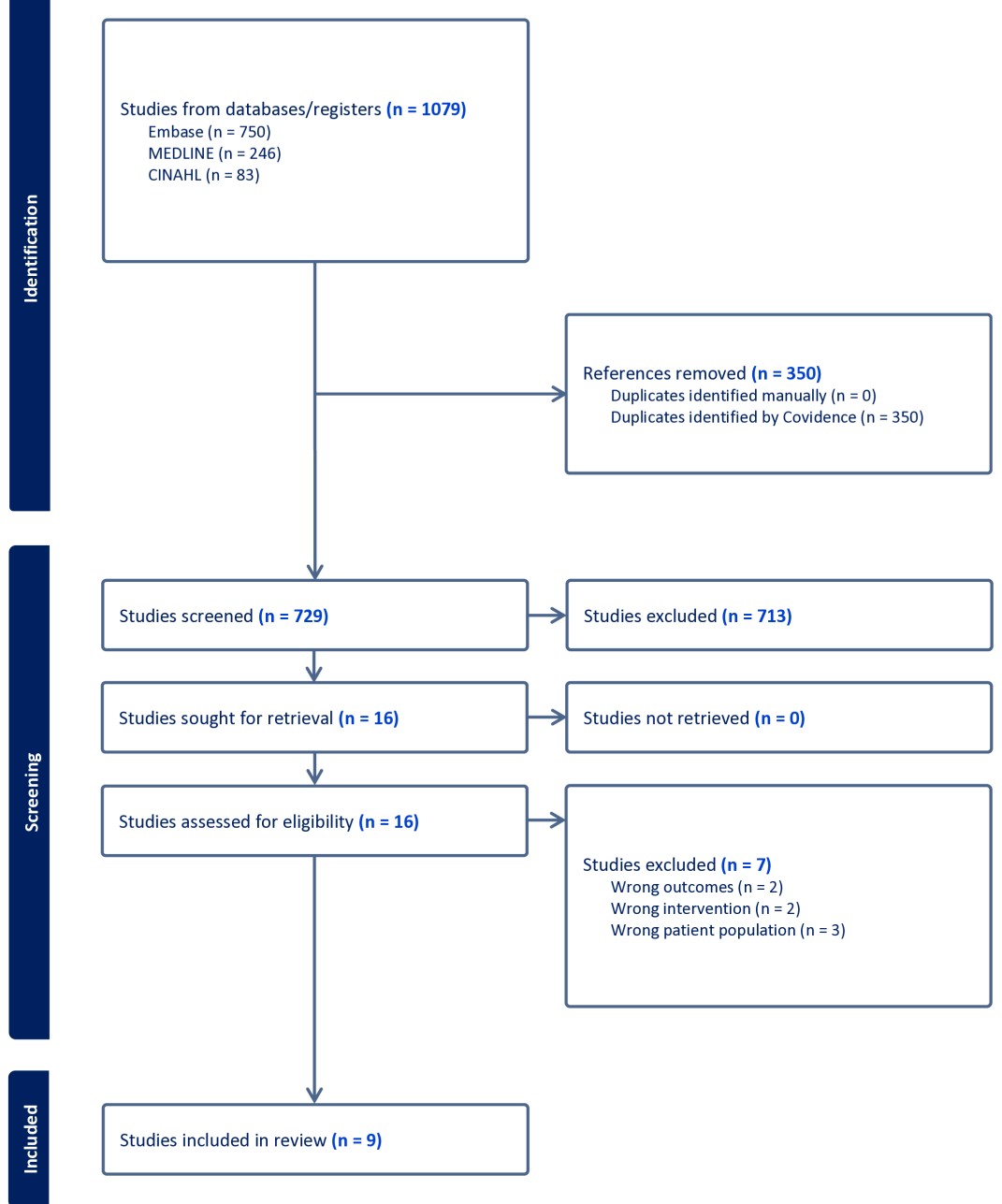

**Fig 1. This PRISMA diagram demonstrates the search, screening, and selection process for the rapid review.** There were no articles added to the rapid review search through citation searing or grey literature searching.

significant reductions in anxiety and depression symptoms, statistically significant improvements in self-esteem, and statistically significant reductions in general health scores immediately following the intervention program [39,40]. A low score in the General Health questionnaire used in this study indicates a favourable psychological state [44,45]. The FIT-PLESE study found statistically insignificant decreases in fertility QoL in both groups after the intervention [42]. The LIFEstyle

**Table 1. Summary of study characteristics included in the rapid scoping review.**

| Author & Date | Age (years) | Sample size | BMI at baseline (kg/m^2) | Cause of infertility | Intervention | Psychological outcomes measured | Results |
|---|---|---|---|---|---|---|---|
| Boiya Rivia et al., 2020 | <40 years | A sample size of 94 has been determined to be required for this study after statistical analysis. | Overweight (>25 kg/m^2) and obese (>30 kg/m^2) participants included. Participants with BMIs >40 kg/m^2 were excluded. | Included: primary infertility. Excluded: bilateral obstruction of fallopian tubes, endometriosis, amenorrhea not due to PCOS, lonely gestational desire, male factor infertility. | Randomized control trial involving a 12-week internet-based lifestyle intervention for women with obesity and infertility in Spain. Lifestyle intervention involves 9 sequential learning modules which promote psychological strategies for making gradual lifestyle changes in diet and physical activity. One interactive component of the program includes the opportunity for participants to log their physical activity hours. | * | * |
| Galletly et al., 1996a | 31 +/- 4.5 years | 64 | 102.3 kg +/- 18.9 kg BMI data not available, however obesity is described as an inclusion criteria for participants. | Cause of fertility not specified, though female participants with infertility are described to be included in the study. | 24-week intervention, and involved one hour of exercise followed by an hour long group session with a psychiatrist, a dietitian, or a reproductive medicine specialist | Anxiety and Depression (Hospital Anxiety and Depression Scale) | Statistically significant improvement in depression scores. A slight reduction was present in anxiety scores. |
| | | | | | | Self-Esteem (Rosenberg Self Esteem Scale) | Statistically significant improvement in self-esteem |
| Galletly et al., 1996b | 31.2 +/- 4.6 years | 37 | 37 +/-6.7 kg/m^2 | Tubal occlusion, male factor infertility, and failure to ovulate. | 24-week intervention, and involved one hour of exercise followed by an hour long group session with a psychiatrist, a dietitian, or a reproductive medicine specialist | Anxiety and Depression (Hospital Anxiety and Depression Scale) | Anxiety and depression scores were significantly reduced after intervention within the study group. |
| | | | | | | Self-Esteem (Rosenberg Self Esteem Scale) | Self esteem significantly rose in score after the intervention within the study group. |
| | | | | | | General Health (General Health Questionnaire) | Significant reduction present in GHQ scores after the intervention within the study group. |
| Legro et al., 2022 | 32.4 +/-4.0 years in control group, 32.1 +/-4.5 years in intervention group | 379 | 39.4 +/-6.9 kg/m^2 in control group, 39.2 +/-7.0 kg/m^2 in intervention group | Participants were included if in good health, met infertility criteria, had regular ovulation, normal ovarian reserve, and normal uterine cavity. Patient must also have at least 1 open fallopian tube, or evidence of an unassisted intrauterine pregnancy within the past 3 year. Male partner must have >5 million motile sperm in ejaculate within 1 year of study. | Both control and experimental group participated in a 12 week exercise-based intervention. Control group participated in increased physical activity alone. Experimental group focused on weight loss through increased physical activity, calorie restriction, and anti-obesity medication (Orlistat) | Fertility-Related QoL | Fertility QoL scores decreased by -0.8 +/-9.0 in control group and -1.6 +/10.2 in intervention group, with a p-value of 0.670. |

*(Continued)*

| Author & Date | Age (years) | Sample size | BMI at baseline (kg/m^2) | Cause of infertility | Intervention | Psychological outcomes measured | Results |
|---|---|---|---|---|---|---|---|
| Mutsaerts et al., 2010 | 18-39 years | A sample size of 520 has been determined to be required for this study after statistical analysis. | >29 kg/m^2 and <40 kg/m^2 | Inclusion: subfertility Exclusion: azoospermia, endometriosis, chronic anovulation class | Multicenter randomized control trial involving a 6-month lifestyle intervention. The LIFEstyle study targeted 5–10% weight loss as a primary objective through a changing participant's dietary pattern, exercise habits, and through a behavioral modification program. | * | * |
| Van Dammen et al., 2018 | 29.7 +/- 4.5 years in the intervention group 29.8 +/- 4.6 years in the control group. | 577 | 36.1 +/- 0.2 kg/m^2 in the intervention group and 36.0 +/- 0.2 kg/m^2 in the control group. | Included: Patients with chronic anovulatory infertility or PCOS Excluded: severe endometriosis, premature ovarian insufficiency. | Multicenter randomized control trial involving a 6-month lifestyle intervention. The LIFEstyle study targeted 5–10% weight loss as a primary through a changing participant's dietary pattern, exercise habits, and through a behavioral modification program. | Physical and Mental QoL (SF-36) | Statistically significant improvement in physical QoLwas evident in the intervention group after the intervention. No difference in mental QoL between control and intervention groups |
| Van Dammen et al., 2019 | 30.0 +/- 4.2 years | 178 | 36.0 +/- 3.2 kg/m^2 | Included: Patients with chronic anovulatory infertility or PCOS Excluded: severe endometriosis, premature ovarian insufficiency. | Multicenter randomized control trial involving a 6-month lifestyle intervention. The LIFEstyle study targeted 5–10% weight loss as a primary through a changing participant's dietary pattern, exercise habits, and through a behavioral modification program. In this article sleep quality, perceived stress, mood symptoms, and QoL are measured in control and intervention group of the LIFEstyle study 5 years post randomization. | Perceived Stress (10-Item Perceived Stress Scale) | No difference in perceived stress between control and intervention groups 5 year post randomization follow up. |
| | | | | | | Anxiety and Depression (Hospital Anxiety and Depression Scale) | No difference in symptoms of depression or anxiety between control and intervention groups 5 year post randomization follow up. |
| | | | | | | Sleep Quality (Pittsburgh Sleep Quality Index) | No difference in sleep quality between control and intervention groups 5 year post randomization follow up. |
| | | | | | | Physical and Mental QoL (SF-36) | No difference in physical or mental QoL between control and intervention groups at 5 year post randomization follow up. |

*(Continued)*

**Table 1.** (Continued)

| Author & Date | Age (years) | Sample size | BMI at baseline (kg/m^2) | Cause of infertility | Intervention | Psychological outcomes measured | Results |
|---|---|---|---|---|---|---|---|
| Wang et al., 2021 | PCOS group 27.9+/-3.9 years, and Non-PCOS group 30.8+/-4.4 years | 259 | PCOS group 35.9+/-3.4 and Non-PCOS group 36.1+/-3.4 | This study is a post hoc longitudinal analysis of the LIFEstyle study. Participants were included in the original study if they were experiencing infertility and excluded if they had severe endometriosis, WHO3 anovulation, endocrinopathies, or previous pregnancy-induced complications. This study investigates participants in the intervention arm of the LIFEstyle study and compares outcomes based on PCOS status. | This study is a post hoc longitudinal analysis of the LIFEstyle study. Multicenter randomized control trial involving a 6-month lifestyle intervention. The LIFEstyle study targeted 5–10% weight loss as a primary through a changing participant's dietary pattern, exercise habits, and through a behavioral modification program. | Physical and Mental QoL (SF-36) | This study is a post hoc longitudinal analysis of the LIFEstyle study. There were no significant differences found in change of mental QoL scores between participants with and without PCOS. Participants with PCOS had a slightly lower physical QoL score at 3 months of intervention compared to non-PCOS participants. |
| Van de Beek et al., 2017 | 18-39 years | 564 WOMB women will be eligible 305 WOMB kids will be eligible for follow up Research team aims to recruit 90 women in both the intervention and control arm of the study. | >=30 kg/m^2 | This is a follow-up of the LIFEstyle study. Participants were included in the original study if they were experiencing infertility and excluded if they had severe endometriosis, WHO3 anovulation, endocrinopathies, or previous pregnancy-induced complications. | This study is a post hoc longitudinal analysis of the LIFEstyle study. Multicenter randomized control trial involving a 6-month lifestyle intervention. The LIFEstyle study targeted 5–10% weight loss as a primary through a changing participant's dietary pattern, exercise habits, and through a behavioral modification program. | * | * |

Study characteristics of nine articles were reviewed and are presented. Three of the articles reviewed presented study protocols only. The lack of available outcome data is indicated by *. QoL = Quality of Life.

study found statistically significant reductions in physical QoL immediately after the intervention in the intervention group and no difference in mental QoL between intervention and control groups [41]. In a follow up study, found no significant difference in any of QoL, anxiety, depression, sleep quality, and perceived stress measures between intervention or control groups at 5 years post randomization [15]. Further, between PCOS and non-PCOS participants, they found no significant difference in mental QoL score change but slightly lower physical QoL scores at the 3-month measurement during the intervention [43].

In general, the research to date suggests that exercise-based interventions may improve psychological measures such as anxiety, depression, perceived stress, self-esteem, general health and physical quality of life (QoL) in the short term. However, positive psychological effects have not been observed past immediate, end-of-study assessments, as documented in Van Dammen's 2019 5-year post-randomization follow-up study [15]. Further, there is no evidence of improvement in mental QoL after engaging in exercise-based interventions for women with obesity and infertility [41]. Exercise

interventions may have less effects on physical QoL in participants with PCOS [43]. Protocols for two prospective studies have been published to investigate the psychosocial effects of exercise-based interventions [36,38].

**Risk of bias.** The table in S2 Text. summarises the risk of bias assessment outcomes for studies included in this rapid review. Two studies had a "high" risk of bias due to self-reporting methods and missing outcome data [39,40]. Three studies had "some concerns" of bias due to self-reporting methods for outcome measurement [15,41,43]. Two studies had a low risk of bias, though both studies were missing information to fully characterise the risk of bias in all domains [36,42]. One study had inadequate information to fully characterise the risk of bias [37]. Further, articles 1, 5, and 9, describe protocol information only and do not comment on the results of the study [36–38]. As a result, the risk of bias could not fully be assessed in these studies.

## Pilot study

Eleven participants were enrolled in the exercise intervention study. Pre-intervention, nine participants completed all of the questionnaires, while two participants completed some questionnaires. Post-intervention, seven participants completed all post-intervention questionnaires. Four participants did not complete the 12-week exercise intervention and dropped out of the study and thus, did not complete the post-intervention questionnaires.

**Participant characteristics.** The participants ranged from 28 to 42 years of age, with an average of $34 \pm 3.7$ at the time of enrolment. Self-reported anthropometric measures (e.g., height and weight) were collected using a survey pre-intervention and after the completion of the intervention. BMI ($kg/m^2$) was calculated before and after the intervention. Pre-intervention participants' (n = 11) average BMI was $40.3 kg/m^2 \pm 4.5$ After completing the exercise program, the average BMI of participants (n = 7) was reduced to $38.9 kg/m^2 \pm 5.8$. Measures of age, height, weight and BMI at pre-intervention and post-intervention are reported in Table 2.

Five surveys were administered to participants to characterize the cohort's psychosocial profiles. Results from pre-intervention and post-intervention survey administration are summarized in Table 3.

**Measures of anxiety and depression.** Individual scores for depression and anxiety were measured using the Hospital Anxiety and Depression Scale (HADS). Pre-intervention (n = 11), participant scores for depression and anxiety averaged to be $8.4 \pm 3.5$ for depression and $4.3 \pm 2.2$ for anxiety, respectively. Seven participants (63.6%) reported having considerable symptoms of depression, and one participant (9.1%) reported considerable symptoms of anxiety. Post-intervention (n = 7), the average score for depression decreased 2.1 points from pre-intervention to $6.3 \pm 3.0$. Scores for anxiety rather increased from pre-intervention to $4.9 \pm 2.5$. After the intervention, the proportion of those experiencing considerable symptoms of depression decreased to 42.9% while the proportion of those experiencing considerable symptoms of anxiety increased to 28.6%. Figs 2 and 3 depict the baseline and post-intervention scores from each participant.

**Table 2. Self-reported demographic and anthropometric data as pre-intervention (n = 11) and post-intervention (n = 7).**

| | | Participant ID (n = 11) | | | | | | | | | | | Mean | SD |
|---|---|---|---|---|---|---|---|---|---|---|---|---|---|---|
| | | 001 | 005 | 006 | 008 | 009 | 013 | 017 | 019 | 024 | 027 | 031 | | |
| Baseline | Age (years) | 33 | 32 | 30 | 36 | 37 | 28 | 34 | 33 | 42 | 36 | 35 | 34 | 3.737 |
| | Weight (kg) | 101.6 | 124.3 | 88.9 | 124.7 | 113.4 | 68.0 | 120.2 | 113.4 | 88.5 | 133.8 | 110.7 | 108.0 | 19.52 |
| | Height (m) | 1.575 | 1.676 | 1.651 | 1.702 | 1.651 | 1.448 | 1.676 | 1.626 | 1.549 | 1.702 | 1.676 | 1.630 | 0.074 |
| | BMI | 41.0 | 44.2 | 32.6 | 43.1 | 41.6 | 32.5 | 42.8 | 42.9 | 36.8 | 46.2 | 39.4 | 40.3 | 4.54 |
| Post-Intervention | Weight (kg) | 102.3 | 124.1 | 82.7 | * | * | 68.0 | 120.2 | * | 88.5 | 129.1 | * | 102.1 | 23.32 |
| | Height (m) | 1.575 | 1.676 | 1.651 | * | * | 1.448 | 1.676 | * | 1.549 | 1.702 | * | 1.630 | 0.091 |
| | BMI | 41.2 | 44.2 | 30.3 | * | * | 32.5 | 42.8 | * | 36.8 | 44.6 | * | 38.9 | 5.78 |

Obesity is defined as having a BMI ($kg/m^2$) equal to or above 30, while severe or morbid obesity is defined as having a BMI ($kg/m^2$) greater than or equal to $40 kg/m^2$. BMI ($kg/m^2$) was calculated by the research team using self-reported measures of weight and height. * No Data.

**Table 3. Baseline and post-intervention psychosocial data.**

| Scale | Timepoint | Parameters | 001 | 005 | 006 | 008 | 009 | 013 | 017 | 019 | 024 | 027 | 031 | Mean | SD |
|---|---|---|---|---|---|---|---|---|---|---|---|---|---|---|---|
| HADS | Baseline | Anxiety | 6 | 5 | 3 | 2 | 2 | 7 | 1 | 8 | 3 | 5 | 3 | | |
| | | Depression | 6 | 9 | 10 | 11 | 4 | 12 | 4 | 14 | 4 | 10 | 8 | | |
| | Post-Intervention | Anxiety | 9 | 7 | 3 | * | * | 5 | 2 | * | 3 | 5 | * | | |
| | | Depression | 7 | 9 | 8 | * | * | 4 | 2 | * | 4 | 10 | * | | |
| Beck Hopelessness Scale | Baseline | | 2 | 6 | 1 | 1 | 2 | 1 | 0 | * | 2 | 8 | 2 | | |
| | Post-Intervention | | 0 | 3 | 1 | * | * | 0 | 0 | * | 2 | 7 | * | | |
| Multidimensional Scale of Perceived Social Support | Baseline | | 84 | 84 | 68 | 84 | 79 | 80 | 84 | * | 82 | 66 | 65 | | |
| | Post-intervention | | 82 | 84 | 62 | * | * | 71 | 82 | * | 81 | 66 | * | | |
| SF-12 (V2) | Baseline | Physical | 46.32 | 54.05 | 43.89 | 51.87 | 54.32 | 51.90 | 53.07 | 56.66 | 46.18 | 50.99 | 56.58 | 51.44 | 4.27 |
| | | Mental | 39.28 | 35.59 | 49.15 | 41.61 | 52.82 | 44.36 | 57.19 | 29.35 | 51.19 | 33.76 | 38.98 | 43.03 | 8.74 |
| | Post-intervention | Physical | 61.83 | 57.78 | 54.85 | * | * | 61.83 | 57.78 | * | 51.58 | 44.69 | * | 55.76 | 6.09 |
| | | Mental | 24.93 | 32.09 | 28.51 | * | * | 24.93 | 32.09 | * | 49.06 | 37.65 | * | 32.75 | 8.48 |
| FPI | Baseline | Social Concern | 31 | 27 | 36 | 27 | * | 34 | 37 | *8 | 35 | 31 | 38 | 32.89 | 4.11 |
| | | Sexual Concern | 21 | 31 | 25 | 27 | * | 39 | 31 | * | 24 | 36 | 26 | 28.89 | 5.86 |
| | | Relationship Concern | 28 | 30 | 30 | 30 | * | 38 | 26 | * | 45 | 33 | 39 | 32.89 | 6.35 |
| | | Rejection of Child Free Lifestyle | 32 | 21 | 31 | 25 | * | 29 | 26 | * | 45 | 33 | 39 | 31.22 | 7.33 |
| | | Need for Parenthood | 35 | 38 | 38 | 29 | * | 30 | 39 | * | 28 | 29 | 31 | 33 | 4.47 |
| | | Fertility-Related Stress Scores | 147 | 147 | 160 | 138 | * | 170 | 159 | * | 161 | 172 | 176 | 158.89 | 12.81 |
| | Post-intervention | Social Concern | 25 | 37 | 33 | * | * | 32 | 28 | * | 30 | 26 | * | 30.14 | 4.22 |
| | | Sexual Concern | 25 | 29 | 28 | * | * | 33 | 24 | * | 25 | 35 | * | 28.43 | 4.24 |
| | | Relationship Concern | 29 | 30 | 30 | * | * | 30 | 26 | * | 27 | 36 | * | 29.71 | 3.20 |
| | | Rejection of Child Free Lifestyle | 32 | 26 | 26 | * | * | 30 | 31 | * | 37 | 37 | * | 31.28 | 4.54 |
| | | Need for Parenthood | 27 | 39 | 47 | * | * | 28 | 36 | * | 28 | 30 | * | 33.14 | 7.46 |
| | | Fertility-Related Stress Scores | 138 | 161 | 164 | * | * | 153 | 145 | * | 147 | 164 | * | 153.14 | 10.25 |

*No Data.

Pre and post-intervention measures of anxiety, depression, hopelessness, perceived social support, quality of life, and fertility-related stress for each participant are captured in Table 3. The validated questionnaires used to obtain these measurements were HADS, Beck Hopelessness Scale, Multidimensional scale of perceived social support, SF-12(V2), and FPI.

**Health-related QoL.** Health-Related QoL was measured using the SF-12v2. Pre-intervention, the sample average pre-intervention for mental and physical components scores was within normal ranges. After the intervention, a general increase was observed in the physical component score to 55.7 (± 6.1) from 51.4 (± 4.3) pre-intervention. All the while, a decrease was observed in the mental component score to 32.8 (± 8.5) from 43.0 (± 8.7) pre-intervention. There was some variability in post-intervention QoL score changes amongst participants. Physical component scores increased post-intervention in four of seven participants, while one remained consistent, and two decreased. Alternatively, mental component scores decreased in four of seven participants, while one remained consistent and two increased. Of note,

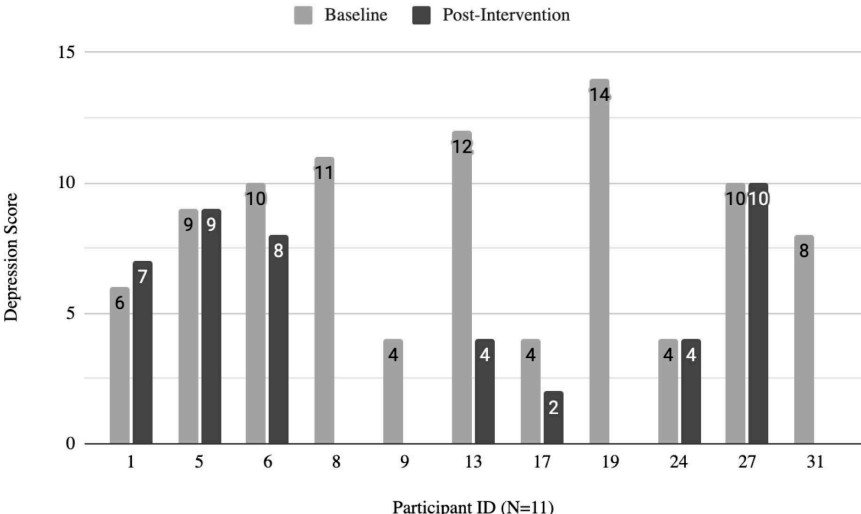

**Fig 2. Participant depression scores at baseline and post-intervention.** Scores range from 0-21. Total subscale scores for depression and anxiety equal to or above eight represent considerable symptoms of depression.

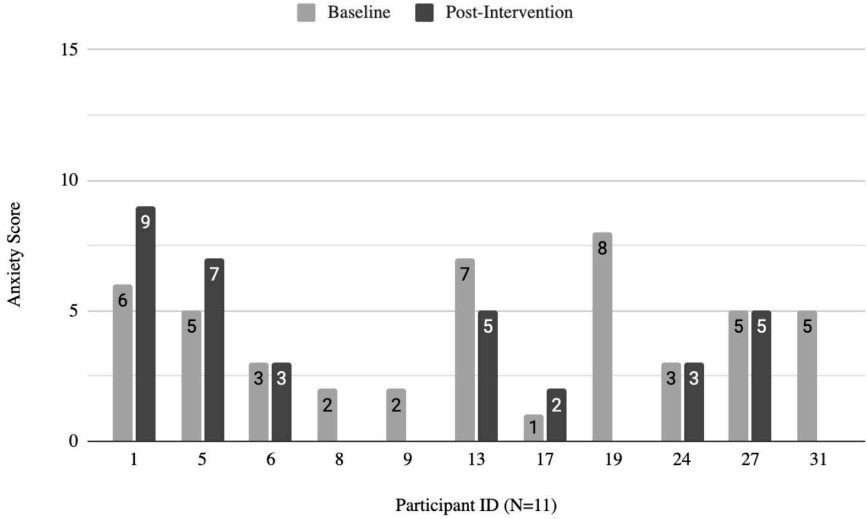

**Fig 3. Participant anxiety scores at baseline and post-intervention.** Scores range from 0-21. Total subscale scores for depression and anxiety equal to or above eight represent considerable symptoms of depression.

one participant's mental component score decreased by over 20 points post-intervention. Table 4 describes the baseline and post-intervention Health-Related QoL scores.

**Infertility-related stress.** Fertility-related stress was measured by using the Fertility Problem Inventory (FPI) [26]. Individual scores for each domain and overall fertility-related stress values are reported in Table 5. Pre-intervention (n = 9), scores for fertility-related stress ranged from 138 to 176, with an average score of 158.9 ± 12.8. Post-intervention (n = 7) fertility-related stress scores ranged from 138 to 164, and the average score reduced to 153.1 ± 10.3. The pre and post-intervention changes measured by the FPI varied by domain. Compared to pre-intervention measures, post-intervention social concern reduced in six out of seven participants (85.7%) and sexual concern decreased in four of

**Table 4. Physical and mental component scores from SF-12v2.**

| | Participant ID (n = 11) | | | | | | | | | | | Mean | SD |
|---|---|---|---|---|---|---|---|---|---|---|---|---|---|
| | 001 | 005 | 006 | 008 | 009 | 013 | 017 | 019 | 024 | 027 | 031 | | |
| Baseline | | | | | | | | | | | | | |
| Physical | 46.32 | 54.05 | 43.89 | 51.87 | 54.32 | 51.90 | 53.07 | 56.66 | 46.18 | 50.99 | 56.58 | 51.44 | 4.27 |
| Mental | 39.28 | 35.59 | 49.15 | 41.61 | 52.82 | 44.36 | 57.19 | 29.35 | 51.19 | 33.76 | 38.98 | 43.03 | 8.74 |
| Post-Intervention | | | | | | | | | | | | | |
| Physical | 61.83 | 57.78 | 54.85 | * | * | 61.83 | 57.78 | * | 51.58 | 44.69 | * | 55.76 | 6.09 |
| Mental | 24.93 | 32.09 | 28.51 | * | * | 24.93 | 32.09 | * | 49.06 | 37.65 | * | 32.75 | 8.48 |

The physical and mental component scores are summary scores based on 8 domains, scores range between 0–100. The summary scores are based on the national average whereby 50 represents an average score for each summary score. A physical component score 50 or less has been recommended as a cut-off to determine a physical condition. While a mental score equal to or below 42 may be indicative of "clinical depression".

*No Data.

**Table 5. Baseline and post-intervention fertility problem inventory data.**

| | Parameters | Participant ID (n = 9) | | | | | | | | | Mean | SD |
|---|---|---|---|---|---|---|---|---|---|---|---|---|
| | | 001 | 005 | 006 | 008 | 013 | 017 | 024 | 027 | 031 | | |
| Baseline | Social Concern | 31 | 27 | 36 | 27 | 34 | 37 | 35 | 31 | 38 | 32.89 | 4.11 |
| | Sexual Concern | 21 | 31 | 25 | 27 | 39 | 31 | 24 | 36 | 26 | 28.89 | 5.86 |
| | Relationship Concern | 28 | 30 | 30 | 30 | 38 | 26 | 29 | 43 | 42 | 32.89 | 6.35 |
| | Rejection of Child Free Lifestyle | 32 | 21 | 31 | 25 | 29 | 26 | 45 | 33 | 39 | 31.22 | 7.33 |
| | Need for Parenthood | 35 | 38 | 38 | 29 | 30 | 39 | 28 | 29 | 31 | 33 | 4.47 |
| | Fertility-Related Stress Scores | 147 | 147 | 160 | 138 | 170 | 159 | 161 | 172 | 176 | 158.89 | 12.81 |
| Post-Intervention | Social Concern | 25 | 37 | 33 | * | 32 | 28 | 30 | 26 | * | 30.14 | 4.22 |
| | Sexual Concern | 25 | 29 | 28 | * | 33 | 24 | 25 | 35 | * | 28.43 | 4.24 |
| | Relationship Concern | 29 | 30 | 30 | * | 30 | 26 | 27 | 36 | * | 29.71 | 3.20 |
| | Rejection of Child Free Lifestyle | 32 | 26 | 26 | * | 30 | 31 | 37 | 37 | * | 31.28 | 4.54 |
| | Need for Parenthood | 27 | 39 | 47 | * | 28 | 36 | 28 | 30 | * | 33.14 | 7.46 |
| | Fertility-Related Stress Scores | 138 | 161 | 164 | * | 153 | 145 | 147 | 164 | * | 153.14 | 10.25 |

*No Data.

seven participants (57.1%). Rejection of a childfree lifestyle increased in four of seven participants (57.1%). Relationship concern remained the same or approximately the same or nearly the same in four of the seven participants (57.1%). Last, for the need for parenthood domain, three of seven (42.9%) reported a need for parenthood. Ultimately, the fertility-related stress scores decreased in five of seven participants (71.4%) post-intervention.

**Hopelessness.** Hopelessness was measured using the Beck Hopelessness Scale (BHS) [22]. Ten of eleven participants completed the questionnaire pre-intervention, and seven completed the questionnaire post-intervention. Of the ten participants pre-intervention, only two participants (18.2%) reported mild hopelessness, with the remaining eight participants scoring (72.7%) in the normal range. Post-intervention, three of seven participant scores (42.9%) remained constant, while four participants (57.1%) reported decreases (a reduction) in hopelessness. The baseline and post-intervention BHS scores are reported in Table 3.

**Perceived social support.** The Multidimensional Scale of Perceived Social Support (MSPSS) was used to measure perceived social support [27]. Pre-intervention, ten of eleven participants completed the questionnaires and post-intervention, seven of seven completed the questionnaire. All participants reported scores signifying high levels of

perceived social support before and after the intervention. Post-intervention, five of the seven participants reported decreases in their scores compared to pre-intervention, and two scores remained constant. Individual data for both time points are displayed in Table 3.

**Patient satisfaction themes.** Three themes emerged from the open-ended responses on the weekly satisfaction survey that highlighted satisfaction with the program and areas where modifications could be made to enhance participant experience. These themes were: (1) external motivation, (2) taking charge and control of adherence, and (3) the need for resources to overcome barriers. Please see Fig 4, which illustrates the themes and sub-themes for participants' attitudes towards the exercise intervention.

Several participants expressed positive feedback about the instructor and how the instructor motivated them during the intervention. The theme of external motivation emerged as crucial for maintaining adherence to the program, but it

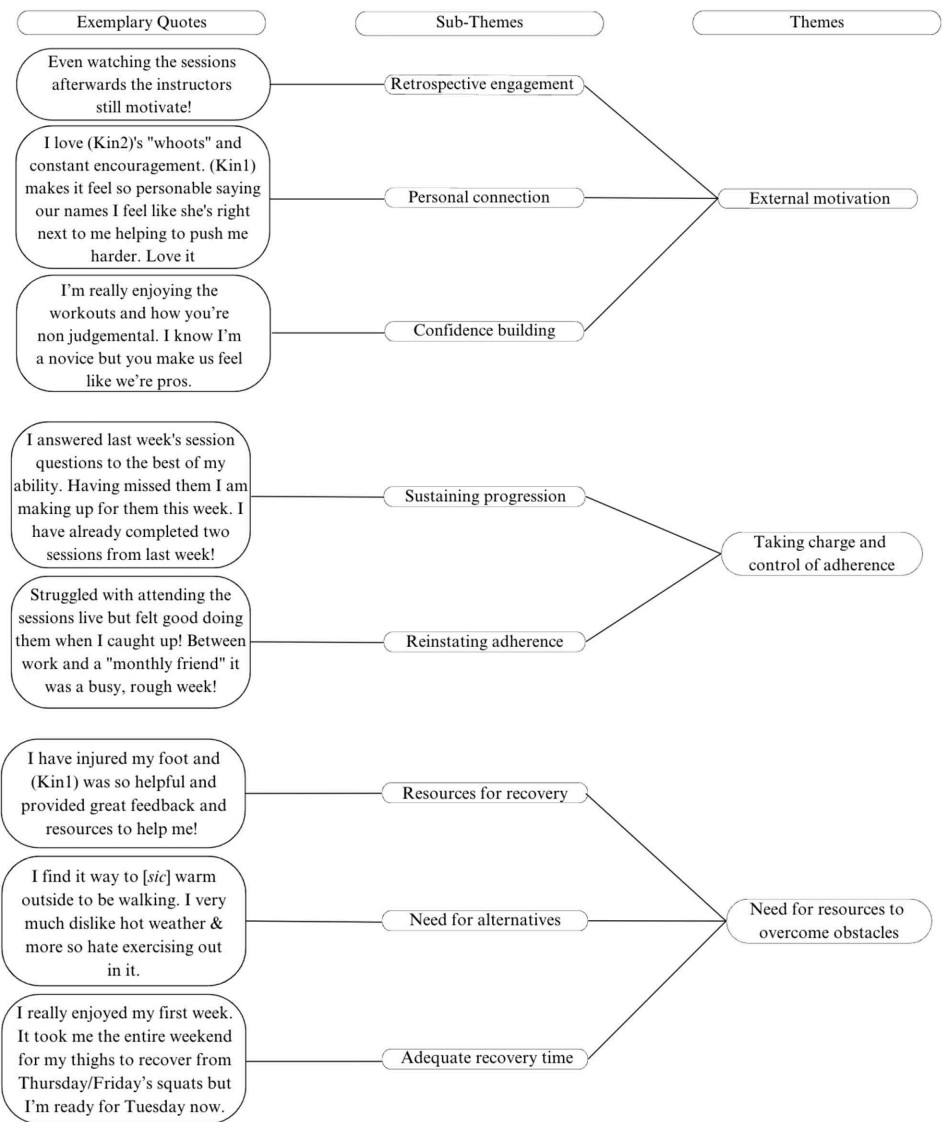

**Fig 4. Themes and sub-themes for participants' attitudes towards the exercise intervention.**

also emphasized the potential need to foster intrinsic motivation. The next theme that emerged from the data was taking charge and control of adherence. For example, after periods of missing sessions, participants noted a desire to get back on track with the program and reach their goals. Having adherence goals set, specific session times and a Facebook group likely provided encouragement to continue with the program after setbacks. The last theme was the need for resources to overcome obstacles. There were multiple quotes from participants that identified resources in place to help participants overcome obstacles, such as advice on injuries and providing adequate recovery time but also highlighted where alternative arrangements could be made, such as considering the time of year and weather conditions.

## Discussion

Our rapid review highlights gaps in the measurement of psychosocial outcomes when assessing the impact of exercise-based interventions for persons with infertility and obesity. The pilot study was designed to engage participants in a moderate-to-high exercise intervention and measure changes in psychosocial outcomes. Overall, general improvements in psychosocial measures following lifestyle interventions were reported in the reviewed studies and our pilot study. However, there were deviations in several psychosocial outcomes in our pilot study compared to studies included in the rapid review. These differences may be attributable to variations in the methodology of the studies. Study-dependent similarities and differences in psychosocial outcomes will be discussed in the following sections.

### Anxiety and depression

Anxiety and depression are commonly reported among persons living with obesity and experiencing female infertility [46–48]. In our study sample of eleven participants, as demonstrated by scores on the HADS questionnaire, more than half of the participants reported considerable symptoms of depression pre-intervention. Studies included in the rapid review suggest that an exercise-based intervention may improve measures of anxiety and depression in participants living with obesity and infertility [39,40]. Thus, based on the literature, we anticipated that self-reported symptoms of depression and anxiety would decrease after the completion of the intervention [39,40,46,49]. In our study sample, the overall improvement in depressive symptoms aligns with the results found in other studies [41,49]. However, the increase in symptoms of anxiety reported by our sample has not been explained by other studies. In the Galletly et al. (1996a) study, authors suggest the less profound effect on anxiety symptoms observed in their study may be related to anticipatory anxiety, as participants would return to fertility treatments once the study had ended [39]. The effect of anticipatory anxiety may have influenced post-intervention scores in the present pilot study as well.

Differences in anxiety and depression-related outcomes between the pilot study and studies included in the rapid review may be due to the inclusion of a counseling component to the exercise-based intervention in the studies included in the rapid review. This difference in intervention concept may have influenced the differences in post-intervention anxiety measures between the pilot study and studies included in the rapid review [39]. Furthermore, it is important to acknowledge the potential impact of the COVID-19 pandemic on outcomes related to mental health in the present pilot study. According to the WHO, the pandemic resulted in a 25% increase in anxiety and depression globally [50]. Levels of depression and anxiety were further increased for patients seeking fertility services during the COVID-19 pandemic [4,50]. Fertility patients were affected by clinic closures without any anticipated re-opening date, leading to delayed or cancelled appointments and procedures [4]. As a result, the effects of the COVID-19 pandemic and suspensions of fertility treatment may have influenced the mental health of participants in the study, confounding post-intervention measurements of anxiety.

### Quality of life

QoL measures are designed to allow patients' perspectives on the impact of health and healthcare interventions on their lives [51–54]. QoL was measured in one study from rapid review using the SF-36 survey [41] and in the pilot study using

the SF-12v2 survey. Both van Dammen et al. and the current study found improvements in physical QoL [41]. However, Van Dammen et al. found no significant change in mental QoL scores while the present study found decreases in mental QoL scores [41]. Similarly to our pilot study, the primary objective of the van Dammen et al. study's intervention was for participants to experience physical improvements (e.g., weight loss, cardiorespiratory fitness) rather than to specifically improve psychosocial markers [41]. According to van Dammen et al., the improvements in physical QoL are aligned with other weight loss trials [41]. The lack of a specific psychological counselling component in the 'LIFEstyle' intervention may explain the lack of improvement in the mental components of QoL. This aspect of intervention design may have influenced the decrease in mental components of QoL score in the pilot study as well.

In the pilot study, comparing the SF-12v2 findings to normative data, despite seven of eleven participants having BMIs greater than 40 kg/m$^2$, the majority of participants reported a higher-than-average QoL for the physical component during the initial assessment (pre-intervention). The high physical component scores were unexpected as obesity has been shown to decrease physical and mental components of QoL [53]. In general, the observation of an increased physical component QoL score and a decrease in the mental component QoL score are supported by findings from rapid review and may be explained by the intervention design.

## Unique psychosocial measures included in the pilot study

**Hopelessness.** Hopelessness is a measure unique to the pilot study and was not measured in studies included in the rapid review. Based on feedback during the patient engagement sessions, PWLE collaborators expressed a feeling of hopelessness during their infertility journey. This feedback led the researchers to measure hopelessness using the Beck Hopelessness Scale [22,31]. From previous literature, it has been shown that the longer a patient experiences infertility, the more significant the feelings of hopelessness experienced [4].

Based on our discussions with PWLE collaborators, we anticipated that participation in an exercise program decreased levels of hopelessness. For example, our PWLE collaborators identified the feeling of hopelessness as a significant experience throughout the discussion sessions. Interestingly, only 20% of our participants in the pilot study reported mild hopelessness while the remaining 80% reported levels of hopelessness within the normal range. This difference from expected findings may be due in part to the measurement tools used to measure hopelessness in our population. The Beck Hopelessness Scale measures general hopelessness and is not specific to fertility-related hopelessness. Therefore, our measurement tool may not have adequately captured the sentiments experienced by our PWLE collaborators. Future research should explore the role of hopelessness in people experiencing infertility, using fertility-specific hopelessness measurement tools. Offering additional support to participants to initiate behavioural change (e.g., increased physical activity levels) and provide a sense of control over their fertility journey is also supported by an abundance of literature [48,55–57]. Post-interventional data supported this notion. While participants reported normal levels of hopelessness pre-intervention, the majority of participants reported decreases in self-reported hopelessness, with few participants experiencing mild hopelessness. During our patient engagement session following the completion of the study, PWLE collaborators vocalized the need for additional resources during the pre-treatment phase of infertility when patients are waiting to see a fertility physician. This finding supports the need for additional resources for individuals seeking fertility services.

**Infertility-related stress.** Infertility-related stress, specifically, was not quantified in studies included in the rapid review. Infertility can considerably impact levels of perceived stress. Infertility-related stress was highlighted as an notable issue affecting PWLE collaborators during patient engagement sessions. The Fertility Problem Inventory (FPI) measures infertility-related stress by summing five domains; the domains consist of: (1) sexual concern, (2) social concern, (3) relationship concern, (4) need for parenthood, and (5) rejection of a childfree lifestyle [36]. The summative score is a fertility-related stress score that indicates the overall infertility-related stress experienced in an individual's life [26]. PWLE collaborators discussed the stress of infertility diagnosis during our patient engagement; thus, researchers selected the FPI to measure infertility-related stress before and after the exercise program.

Amongst our participants, 71% reported lower fertility-related stress scores after completing the exercise program. Of interest to the fertility-related stress score finding, the domains of 'Social Concern' and 'Relationship Concern' saw reductions in the overall average post-intervention scores compared to pre-intervention. However, the reduction in relationship concerns is not what we had hypothesized. Previous research has shown that a weight-loss attempt by one partner in a relationship increases negative interactions, threatens the other partner's security, or changes the nature of the relationship [58]. Dailey et al. (2018) observed that weight loss attempts by one or both partners cause imbalance or dissatisfaction in the relationship as measured by using thematic analysis of participant interviews [59]. Researchers found that if individuals had the necessary support from their partners and could rely on their partners to make them healthy food, for example, they would have more time to devote to other responsibilities, ultimately leading to improved relationship satisfaction and successful weight loss [59]. In the present study, reduced relationship concerns for several participants may indicate increased partner support for the intervention.

**Perceived social support.** Discussions from the patient engagement sessions had a significant focus on the importance of social support, specifically surrounding the social support that was provided by other participants in the exercise program. In the pilot study, results from pre- and post-intervention showed all participants reported high levels of social support. Unfortunately, participants reported reduced perceived social support after the intervention's completion, though no changes resulted in a categorical shift (e.g., from high to medium social support). The lack of social support may reflect the virtual aspect of the exercise program or the temporal proximity of the intervention to the COVID-19 pandemic.

**Patient satisfaction.** The analysis of open-ended responses revealed three themes about participants' views on the exercise intervention. One theme, "external motivation," highlights how participants saw Kinesiologists as key motivators for adhering to the exercise program. This aligns with other studies, including a systematic review, which found that supervised exercise enhanced motivation and adherence more than unsupervised exercise [60]. Interestingly, Hu et al. (2022) compared the effects of online supervised exercise to self-directed exercise in an obese population and showed comparable adherence outcomes [61]. Authors concluded that internal motivation may be necessary in an online environment, even in the presence of supervision [62]. Building upon discussions of sources of motivation, the second theme was 'taking charge and control of adherence.' After noting absent periods, participants documented being motivated to "get back on track" with the study. Further, participants reported adherence-based goals, accountability to 'live' session times (e.g., Tuesday and Thursday at 6:00 am), and engagement in the Facebook group were sources of motivation to continue the intervention following setbacks [63]. The third theme that emerged was the 'need for resources to overcome barriers.' Participants highlighted barriers to participation and the desire to overcome these barriers with additional resources. For example, participants discussed that the scheduling of the exercise intervention during the summer months (e.g., taking annual leave during the summer months) and weather conditions (e.g., hot temperatures and lack of air conditioning in the summer months) impacted their ability to adhere to the exercise intervention. In a recent review by Hunter et al. (2021) of randomised controlled trials that utilised physical activity in weight loss interventions for persons living with obesity and experiencing infertility, additional barriers to exercise adherence reported were related to the safety of the neighbourhood and long working hours [62]. For persons with a diagnosis of infertility, supervised exercise, sources of motivation and barriers related to logistics, resources and societal factors must be considered when developing and implementing an exercise intervention for this population.

## Strengths and weaknesses

The present paper exhibits notable strengths. This paper combines information from a pre-experimental pilot study and a rapid scoping review. The rapid scoping review provides context for interpreting results from the pilot study. Next, the pilot study adopted a patient-centered approach, involving patients of an underserved demographic—women with obesity and infertility. Of note, patient-oriented research methods were employed, giving voice to individuals with lived experiences

who contributed significantly to study design and outcomes. The research team also conducted in-person engagement sessions with patients to discuss study objectives and methodology, and validated questionnaires were selected based on the priorities of these PWLE collaborators. Additionally, the use of a virtual study environment was advantageous as it removed participation barriers and reduced the stigma experienced by individuals with obesity. Lastly, the pilot study was not hypothesis-driven, rather, we focused on descriptively reporting our findings. There are many reasons to complete a pilot study, including process, resources, management and scientific reasons [64]. Beyond our description of our findings, we learned that recruitment rates and participant retention rates are low in this population. Therefore, a multiple-site, pan-Canadian trial is necessary on a larger scale. Based on low participant retention rates and the descriptive findings of our psychosocial outcomes, we learned from a resource perspective, behavioural change counselling is important to include in an exercise intervention study for this population.

The rapid scoping review portion of this paper has significant limitations. First, the outcome was included in the search strategy, which limited the articles to be reviewed in the study. Next, there was only one reviewer when screening the articles for inclusion and conference abstracts were excluded from the database searches. Further, only three databases were used for the search and citation searching and grey literature searching were not included in the rapid review. The articles included in the rapid review had either moderate to high risk of bias or were missing adequate information to assess bias. Because of the risk of bias in included articles, our confidence in the rapid review results is limited. Last, the articles included in the rapid review were not critically appraised. This further limits our confidence in the presented results. Unfortunately, these limitations narrowed the scope and reliability of the review.

This pilot study also had limitations. The sample size of our pilot study is small. The objective of our pilot study was to perform a small-scale test of our novel methodological approach (e.g., patient-oriented approach and virtual exercise intervention) to inform a larger-scale study [64]. The sample size was not sufficiently powered for statistical testing, limiting the analysis to descriptive statistics. The virtual nature of the study made it challenging to assess participant adherence and relied heavily on self-reported data. Recruitment hurdles were encountered, necessitating the use of social media to supplement physician referrals. A substantial dropout rate of 36.3% was observed, potentially linked to low self-esteem and the absence of psychological support. The administration of questionnaires in a virtual format faced difficulties, resulting in incomplete or improperly completed forms. Further limitations included our recruitment strategy. For our pilot study, participants started the exercise intervention at the same time, forming a single cohort. In contrast to a rolling recruitment strategy, where participants begin at different times, our approach ensured our participant cohort experienced social support and the program simultaneously from the outset. However, while this strategy was encouraged by our PWLE collaborators, it made recruitment limited as participants were restricted to a certain start and end date. Additionally, a combination of patient-oriented research and quantitative methods extended the study duration, requiring a considerable time commitment from PWLE collaborators and researchers. It is worth noting that while PWLE collaborators were involved in study design, the selection of questionnaires was not confirmed with them and relied on the research team's judgement.

### Future directions and clinical implications

Results from our rapid review and the inclusion of novel measurements and findings from our thematic and descriptive analyses highlight the need for a critical understanding of how exercise affects our patient population. Furthermore, the unexpected increase in anxiety symptoms and the decrease in QoL in mental health observed in our pilot study suggests the need for continued exploration of the impact of exercise prescription for persons with infertility. Clinicians should be cautious in expecting exercise to universally enhance mental health outcomes and thus, consider a more holistic approach that includes psychological counselling or other forms of mental health support. For example, the most commonly cited behavioural change counselling approach for persons with infertility and obesity is motivational interviewing [59]. Motivational interviewing, which encourages behavioural change by applying a patient-centred counselling style, has been implemented successfully to aid in weight loss before fertility treatment [65]. Lifestyle psychiatry, a newer form of

behavioral change counselling that promotes exercise for mental health, has shown positive physical and psychological outcomes [66]. However, it has not yet been studied in people with infertility and obesity. Integrating lifestyle psychiatry in reproductive medicine could improve both QoL and modifiable risk factors linked to infertility.

Lastly, understanding the long-term effects of lifestyle interventions, including exercise-based programs, on psychosocial outcomes for individuals with infertility is crucial. In our pilot study, a reduction in self-reported anxiety and mental QoL from baseline suggests that time may significantly impact these outcomes [8]. Studies by Van Dammen et al. (2019) and Van de Beek et al. (2017) found that the benefits observed immediately after the intervention diminished over time, with no significant differences in QoL, perceived stress, or sleep quality at five years [15,38]. The lack of sustained improvement in psychosocial measures for individuals with obesity and infertility underscores the need for revising intervention designs to enhance their long-term effectiveness. Overall, future research should focus on integrating psychological support within exercise interventions and exploring other factors that contribute to the long-term psychosocial well-being of this population.

## Conclusion

Findings from this rapid review and pilot study suggest that while exercise-based interventions can have a positive impact on certain psychosocial outcomes, exercise alone may not fully address the psychological needs of individuals with obesity and infertility. To maximize the benefits of exercise programs, evidence from the current study suggests that clinicians should incorporate additional support mechanisms, particularly to manage anxiety and improve mental quality of life. Furthermore, our study offers new insights by examining the effects of exercise on hopelessness, perceived social support, and infertility-related stress, all of which showed favourable outcomes. In conclusion, future research should focus on integrating psychological support within exercise interventions and exploring other factors that contribute to the psychosocial well-being of this population.

## Supporting information

**S1 Text. Search strategy for rapid scoping review.** Detailed search strategy sample for rapid scoping review, as applied to Medline search.
(DOCX)

**S2 Text. Rapid scoping review risk of bias assessment.** Cochrane risk of bias assessment figure for rapid scoping review [67]. Bias assessment judgments are summarized for articles included in the rapid scoping review through the Cochrane risk of bias assessment tool. Articles 1, 5, and 9 described study protocols only, hence many bias domains were not able to be assessed due to lack of information.
(DOCX)

**S1 Checklist. PRISMA scoping review checklist.** The PRISMA scoping review checklist summarizes key procedural elements of the present rapid scoping review [68].
(PDF)

## Acknowledgments

We thank our PLWE collaborators for sharing their time, insight, and stories with the research team. We would also like to thank all of the participants of the pilot study for their time and engagement.

## Author contributions

**Conceptualization:** Jillian Murdoch, Deanna Murphy, Erin McGowan, Laurie K. Twells, Katie Wadden.

**Data curation:** Tiffany Furneaux, Jillian Murdoch, Nicole Hollohan, Laurie K. Twells, Katie Wadden.

**Funding acquisition:** Erin McGowan.

**Investigation:** Tiffany Furneaux, Jillian Murdoch, Erin McGowan.

**Methodology:** Alison Farrell, Erin McGowan, Laurie K. Twells, Katie Wadden.

**Project administration:** Tiffany Furneaux, Jillian Murdoch.

**Supervision:** Deanna Murphy, Erin McGowan, Laurie K. Twells, Katie Wadden.

**Writing – original draft:** Tiffany Furneaux, Jillian Murdoch.

**Writing – review & editing:** Tiffany Furneaux, Jillian Murdoch, Nicole Hollohan, Catherine M. Barrett, Deanna Murphy, Alison Farrell, Laurie K. Twells, Katie Wadden.

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
