## [Decision Letter · Decision Letter 0]

12 Aug 2024

PMEN-D-24-00233

The Psychosocial Impact of Exercise as an Intervention for Persons Living With Obesity and Female Infertility: A Rapid Scoping Review and Pilot Study

PLOS Mental Health

Dear Dr. Wadden,

Thank you for submitting your manuscript to PLOS Mental Health. After careful consideration, we feel that it has merit but does not fully meet PLOS Mental Health’s publication criteria as it currently stands. Therefore, we invite you to submit a revised version of the manuscript that addresses the points raised during the review process.

We look forward to receiving your revised manuscript.

Kind regards,

Martin Mabunda Baluku, Ph.D.

Academic Editor

PLOS Mental Health

Journal Requirements:

1. In the online submission form, you indicated that "The data that support the findings of this study are available on request from the corresponding author, KW". 

a. In a public repository, 

b. Within the manuscript itself, or 

c. Uploaded as supplementary information.

Additional Editor Comments (if provided):

Reviewers' comments:

Reviewer's Responses to Questions

**Comments to the Author**

1. Does this manuscript meet PLOS Mental Health’s publication criteria?

Reviewer #1: No

Reviewer #2: Yes

Reviewer #3: Partly

Reviewer #4: Yes

2. Has the statistical analysis been performed appropriately and rigorously?

Reviewer #1: No

Reviewer #2: Yes

Reviewer #3: No

Reviewer #4: Yes

3. Have the authors made all data underlying the findings in their manuscript fully available (please refer to the Data Availability Statement at the start of the manuscript PDF file)?

Reviewer #1: Yes

Reviewer #2: Yes

Reviewer #3: No

Reviewer #4: Yes

4. Is the manuscript presented in an intelligible fashion and written in standard English?

Reviewer #1: Yes

Reviewer #2: Yes

Reviewer #3: Yes

Reviewer #4: Yes

Reviewer #1: This is a nicely written paper on a very interesting topic. An "n" of 7 is just not enough, especially given confounders like the effect of COVID-19. The review is good, and would be a nice paper on its own. You should address important topics like a rating of how much the participants enjoyed the exercise. There are many confounding variables which might explain your findings. I strongly encourage you to continue your student and report findings on a larger N in a post-COVID era. The role of Lifestyle Psychiatry in this unique population is a very important topic.

Reviewer #2: Reviewers’ comments

This is a well-written manuscript, addresses an interesting topic and makes a relevant contribution to the field of both Obesity and Female Infertility. The title is adequate and the abstract is sufficiently addressed. However, few minor revisions are needed before it can be published. The following recommendations are provided:

Background

1.The background provides a clear and structured overview of the problem, linking infertility, obesity, and the psychosocial challenges involved. However, the flow could be improved to ensure smooth transitions between ideas.

2.The paragraphs are long and dense, making it difficult to follow the narrative. Breaking them into shorter, more focused paragraphs would improve readability and comprehension.

3.There is a lack of smooth transitions between different sections and ideas. Adding transitional phrases or sentences would help guide the reader through the text more effectively.

Current Study and Objectives:

4.The dual approach (rapid scoping review and pre-experimental study) is a strength of the study design. Ensure that the rationale for choosing these methods is clear to the reader.

5.The specific measures to be assessed (anxiety, depression, hopelessness, infertility-related stress, and perceived social support) are relevant and comprehensive. It might be helpful to briefly mention why these particular measures were chosen.

Methodology

6.The rapid scoping review did not include a critical appraisal of the articles due to time constraints. This omission could affect the reliability and validity of the findings since the quality of the included studies is not assessed.

7.Both article screening and data extraction were conducted by a single author (JM). This approach could introduce bias and errors in the review process. Involving multiple reviewers in screening and data extraction would enhance the reliability and reduce the potential for individual bias, ensuring a more accurate and comprehensive review.

Discussion

8.The discussion acknowledges deviations in psychosocial outcomes due to methodological variations between the pilot study and those in the rapid review. However, it fails to delve deeply into what specific methodological differences might have caused these variations. More detailed comparison and critical evaluation of methodologies, such as differences in intervention intensity, duration, and participant characteristics, would provide clearer insights into the observed discrepancies.

9.he manuscript starts off strong, but the authors' voice seems to weaken towards the end. The Discussion or Overview section should clearly connect to the study's aim, presenting a broad overview of the results before delving into a more analytical exploration of the implications. Instead, the authors reiterate the studies previously discussed without providing new insights. What is significant about the findings of this review? What should clinicians keep in mind when working with these diagnoses? If additional professional training is necessary, what specific areas do the authors recommend? The lack of a clear takeaway appears to stem from an unclear aim within the manuscript.

10.While the discussion section provides a comprehensive overview of the study's findings and their context within existing literature, it could be improved by addressing the limitations of the study's design and data collection methods more critically.

Reviewer #3: Persons living with obesity and experiencing infertility are counselled on healthy lifestyle behaviours to improve fertility-related outcomes. However, due to the known psychological burden of receiving an infertility diagnosis, there is an important need to consider the psychosocial impact of lifestyle interventions for this population. Studies uses two research methods�that is 1) a rapid scoping 29 review, and 2) a pilot study of a 12-week virtual exercise program�I think the research objective and method is feasible There are two sections that need to be improvement

1. Please provide specific statistical methods

2. Please include specific exercise intervention methods

Reviewer #4: Dear authors, thank you for submitting your manuscript to PLOS Mental Health. Kindly revise the following comments:

For the Abstract Section:

The abstract effectively summarises a complex study, highlighting both its strengths and the areas needing further exploration. Refining the objective statement, clarifying the results, and providing a brief interpretation of the pilot study findings would enhance the clarity and impact of the abstract.

Strengths:

Clear Research Focus: The abstract clearly outlines the dual focus of the research—both the rapid scoping review and the pilot study. This dual approach is well-articulated, demonstrating a comprehensive attempt to understand the psychosocial impacts of exercise interventions on individuals with obesity and infertility.

Structure and Organisation: The abstract follows a logical structure, beginning with the problem statement, followed by the research methods, and ending with the findings. This helps the reader quickly grasp the study's purpose, methods, and key outcomes.

Inclusion of Key Findings: The abstract effectively summarises the findings from both the rapid scoping review and the pilot study, providing specific details on the outcomes measured and the changes observed in psychological measures. The presentation of quantitative data (e.g., number of participants, age, BMI) adds to the clarity and precision of the summary.

Identification of the Need for Further Research: The abstract concludes by acknowledging the unexpected findings in the pilot study, particularly the increase in poor mental health following the exercise programme. This highlights the need for further investigation, which is a crucial element in research abstracts.

Areas for improvement:

Objective Statement: The objective of the study, as stated in lines 28–29, could be more explicitly formulated. The current phrasing is somewhat fragmented and would benefit from a complete sentence that clearly defines the research objectives. For example, “This study aims to assess the psychosocial impact of exercise interventions on individuals with obesity and infertility through a rapid scoping review and a pilot study.”

Methodology Details: While the abstract mentions the use of MedLine, Embase, and CINAHL for the rapid scoping review, it does not provide details about the inclusion/exclusion criteria or the specific search terms used. Including a brief mention of these criteria would enhance the transparency of the methodology.

Clarification of Results: The results section could be more concise and focused. For instance, the presentation of the pilot study findings includes multiple mentions of improvements and declines in different measures, which could be confusing. Grouping the outcomes into broader categories (e.g., improvements in psychological well-being, declines in mental health) could make the results clearer and easier to follow.

Interpretation of Pilot Study Findings: The abstract notes an increase in poor mental health following the exercise programme, which contrasts with the general trend observed in the rapid scoping review. However, it does not provide any interpretation or possible explanation for this discrepancy. A brief mention of potential reasons for these findings or the context of the COVID-19 pandemic’s impact could provide valuable context.

Keywords: The keywords provided are relevant but could be expanded to include terms like “psychosocial outcomes,” “pilot study,” or “exercise intervention” to improve searchability and alignment with the study’s focus.

2. For the Background Section:

Overall, the background is well-crafted, addressing a significant issue with clear research objectives. Enhancing the integration of literature, improving conceptual flow, and providing additional clarity on specific terms and populations would further strengthen the manuscript.

Strengths:

Clarity and Focus: The background provides a clear and focused introduction to the problem of infertility, particularly among individuals living with obesity. The definition of infertility is straightforward, and the statistics on its prevalence in Canada effectively underscore the significance of the issue.

Identification of the Research Gap: The manuscript identifies a relevant research gap, specifically the lack of studies on the psychosocial changes in individuals with obesity and infertility who are prescribed lifestyle interventions. This is well-articulated and establishes the need for the current study.

Integration of Patient Perspective: The inclusion of patient experiences, as derived from discussion groups, strengthens the background. It provides a real-world context that supports the research's relevance and the urgency of addressing the identified gap.

Connection to Existing Research: The manuscript successfully connects the current study to existing research, particularly in highlighting the consequences of inadequate obesity management support for individuals with infertility. This connection is critical for justifying the study’s objectives.

Areas for improvement:

Literature Support: While the background cites relevant literature, it would benefit from a more comprehensive discussion of existing research on the psychosocial effects of infertility and obesity, particularly in the context of prescribed lifestyle interventions. This could help to further emphasise the novelty and importance of the study.

Conceptual Flow: The transition from discussing infertility and obesity to the introduction of the study's objectives could be smoother. For instance, the discussion of physical activity and its relevance to cardiometabolic health (lines 94–98) feels somewhat abrupt and could be better integrated into the overall narrative. A stronger bridge connecting the general background to the study's specific focus on exercise-based interventions would improve the flow.

Specificity of Objectives: While the background clearly outlines the primary objective, it might benefit from more precise wording regarding the methodology. The term "rapid scoping review" (line 101) is introduced without prior explanation. Clarifying what this entails and why it’s appropriate for the research question would enhance the reader's understanding.

Clarification of Patient Population: The manuscript refers to "persons living with obesity and experiencing female infertility" (lines 103–106). It may be helpful to specify the population's demographic characteristics (e.g., age range, severity of obesity) to give a clearer picture of the study’s focus.

Reduction of Redundancy: Some sentences could be streamlined to avoid redundancy. For example, the statement about the psychosocial consequences of infertility (lines 74–76) could be more concise, as it is closely followed by another discussion of the psychosocial outcomes targeted by the study (lines 92–93).

For the methods section:

The methods section is well-structured and provides a comprehensive overview of the research process. Enhancing clarity, providing more justification for methodological choices, and addressing limitations related to sample size and critical appraisal would strengthen this section further. These improvements would ensure that the study’s methodology is not only rigorous but also transparent and replicable.

Strengths:

Comprehensive Detail: The methods section provides a thorough and detailed account of both the rapid scoping review and the pilot study. This level of detail enhances the reproducibility of the research, as it allows other researchers to understand exactly how the study was conducted.

Use of Established Tools and Software: The manuscript demonstrates a solid methodological foundation by using established tools like Covidence software for managing the literature review process and validated questionnaires for measuring psychosocial outcomes. The mention of specific software (e.g., PRO CoRE for SF-12v2 scoring) adds credibility to the data analysis process.

Patient-Orientated Research (POR) Approach: The inclusion of patient partners in the study design is a notable strength. It reflects a commitment to patient-centred research and ensures that the study addresses the real-world concerns of those living with obesity and infertility. The iterative process of consulting with patient partners and modifying the study protocol based on their feedback is well documented.

Adaptability in the Context of COVID-19: The methods reflect a well-considered adaptation to the challenges posed by the COVID-19 pandemic, with the exercise intervention being delivered online. The use of Zoom and the creation of a private Facebook page demonstrate an effort to maintain participant engagement and support despite physical distancing requirements.

Areas for improvement:

Clarity and Consistency in Reporting: The timeline for the rapid scoping review initiation is unclear (lines 112-113: “June/August 2023”). It is essential to specify the exact start date for clarity. Additionally, the methods for identifying key articles during the initial search (line 118) could be more explicitly detailed, including whether any particular filtering or screening process was used to ensure comprehensiveness.

Critical Appraisal Omission: The decision not to critically appraise articles due to time constraints (lines 123–124) is a limitation of the rapid scoping review. This could be addressed more explicitly by acknowledging the potential impact this might have on the findings. The authors should consider justifying this omission more robustly or suggesting how it could be addressed in future studies.

Sample Size and Recruitment Challenges: The recruitment process is described in detail, but the reasons for participant dropout (lines 163–167) could be elaborated further. Understanding the challenges of recruiting and retaining participants, especially in a small pilot study, is crucial. This could include a discussion of strategies that could be employed in future research to enhance recruitment and retention.

Inclusion/Exclusion Criteria: The criteria for participant selection are clear, but the rationale behind some of the exclusion criteria, particularly regarding metabolic disease and specific infertility diagnoses (lines 173–177), could be expanded upon. Explaining why these particular conditions were excluded would add depth to the methodological rationale.

Data Collection and Analysis: While the data collection methods are well documented, the section could benefit from a brief discussion of how missing data (e.g., from participants who did not complete the post-intervention questionnaires) were handled. Additionally, given the small sample size, the limitations of the statistical analysis (limited to descriptive statistics) should be acknowledged more directly, including any potential implications for the interpretation of the results.

For the results section:

The results section is solid but would benefit from additional analysis and discussion. Enhancing clarity, expanding on the statistical analysis, and interpreting the findings in a broader context would significantly strengthen the manuscript.

1. Clarity and Structure:

The results section is well organised, presenting the scoping review, study characteristics, and pilot study outcomes in a logical sequence. However, there is a need for clearer transitions between the different parts of the section. Explicitly demarcating the scoping review results and pilot study findings with headings could enhance readability.

2. Methodological Rigour:

The rapid scoping review process is described adequately, but the exclusion criteria for full-text screening and reasons for article exclusion should be more detailed to ensure transparency and reproducibility.

The PRISMA diagram is referenced but not provided, which limits the ability to assess the thoroughness of the screening process. Including this figure is essential.

3. Data Presentation:

The summary of the studies in Table 2 is comprehensive but can be overwhelming due to its dense format. Consider splitting the table into multiple parts or providing a more condensed summary focussing on key points.

Some studies reported significant outcomes while others did not, but there is no discussion or interpretation of these mixed results. It would be helpful to provide an analysis of why certain studies showed significant outcomes and others did not.

4. Pilot Study Results:

The results of the pilot study are intriguing, but the sample size is small and there is a significant dropout rate. These limitations should be more explicitly acknowledged, as they impact the generalisability and robustness of the findings.

The change in BMI post-intervention is mentioned, but the clinical relevance of this reduction is not discussed. Including a discussion on whether this change is meaningful in the context of infertility and obesity would be valuable.

5. Statistical Analysis:

There is a lack of detailed statistical analysis in the results of the pilot study. The manuscript would benefit from a description of the statistical tests used to assess pre- and post-intervention changes, along with p-values or confidence intervals.

6. Interpretation and Implications:

The conclusion that exercise-based interventions may improve psychological outcomes in the short term is reasonable, but the absence of long-term effects should be discussed in more depth. The manuscript should explore potential reasons for this decline in effect over time.

The mixed results on physical and mental QoL, particularly in the LIFEstyle study, deserve more thorough discussion. The implications of these findings for clinical practice and future research should be considered.

7. Minor Points:

Ensure consistent referencing of studies in the text and tables (e.g., "Galletly et al., 1996a" vs. "Galletly et al., 1996b").

There is a minor typographical error in the sentence on line 227: "searing" should be "searching."

For the discussion section:

Strengths of the Study Design and Integration with Literature:

The discussion effectively connects the study’s aim to assess exercise-based interventions on psychosocial outcomes in individuals with infertility and obesity with previous research. However, the section on variations in psychosocial outcomes could be strengthened by exploring specific factors like sample characteristics, intervention duration, or measurement tools that might explain these deviations.

Addressing Confounding Factors:

The consideration of the COVID-19 pandemic as a confounding factor is commendable and adds validity to the study. To further improve, the discussion could provide more detailed insights into how the pandemic specifically affected the study participants, rather than relying solely on global statistics.

Consistency and coherence:

The discussion is well-structured, with clear subsections. However, smoother transitions between sections, such as between anxiety, depression, and quality of life, would enhance coherence.

Consideration of Unique Measures:

The inclusion of unique psychosocial measures, like hopelessness and infertility-related stress, adds depth to the study. The discussion could be improved by providing context on why hopelessness was not included in previous studies and what its inclusion reveals about the studied population.

Strengths and Weaknesses of the Study:

The strengths and weaknesses are discussed candidly. However, while the limitations are thoroughly covered, the strengths, particularly the patient-centred approach and integration of patient feedback, could be emphasised more to highlight the study’s significance.

Conclusion and Future Directions:

The discussion could benefit from a more explicit conclusion that summarises the key findings and their implications, along with specific recommendations for future research or clinical practice, such as incorporating psychological support into exercise-based interventions.

Recommendations:

Provide a detailed analysis of why certain psychosocial outcomes differed from previous studies.

Offer more context on how the COVID-19 pandemic specifically impacted the study participants.

Improve transitions between discussion sections for better coherence.

Expand on the significance of including hopelessness as a measure in the context of infertility and obesity.

Balance the discussion of limitations with a stronger emphasis on the study’s strengths.

Conclude with a clear summary of key findings and practical recommendations for future research and interventions.

For the conclusion section:

Clarity and conciseness:

The conclusion is clear and effectively summarises the key findings. However, it could benefit from a more structured approach, separating the summary of findings from the implications.

Integration of Findings:

The integration of rapid review and pilot study findings is well done. To strengthen this section, the authors could more explicitly compare and contrast these findings.

Discussion of Implications:

The conclusion touches on potential explanations for the observed effects, such as infertility-related distress and the impact of COVID-19. Expanding on the implications for future research or practice would enhance this section.

Strength of Statements:

The conclusion effectively supports the findings but could avoid broad generalisations. For example, the statement on mental quality of life could be softened to reflect context-dependent factors.

Future Directions:

Adding a sentence on future research directions or practical implications would provide a forward-looking perspective.

Finality and Impact:

The conclusion could end with a stronger, more impactful statement, emphasising the importance of addressing both physical and psychosocial outcomes in interventions.

Confidential Recommendations:

Restructure the conclusion for clarity, separating findings from implications.

Compare pilot study findings with those from the rapid review to highlight contributions.

Expand on the implications for future research and practice.

Soften generalisations to reflect context-dependence.

Add a forward-looking perspective, suggesting future research directions or practical applications.

End with a strong, impactful statement.

**Do you want your identity to be public for this peer review?** For information about this choice, including consent withdrawal, please see our Privacy Policy

Reviewer #1: No

Reviewer #2: **Yes: ** Dr. Jesan Ara

Reviewer #3: No

Reviewer #4: **Yes: ** Ayah Talal Zaidalkilani

---

## [Decision Letter · Decision Letter 1]

3 Apr 2025

PMEN-D-24-00233R1

The Psychosocial Impact of Exercise as an Intervention for Persons Living With Obesity and Female Infertility: A Rapid Scoping Review and Pilot Study

PLOS Mental Health

Dear Dr. Wadden,

Thank you for submitting your manuscript to PLOS Mental Health. After careful consideration, we feel that it has merit but does not fully meet PLOS Mental Health’s publication criteria as it currently stands. Therefore, we invite you to submit a revised version of the manuscript that addresses the points raised during the review process.

Please attend to the requests for clarity and greater depth of discussion made by Reviewer 5. Furthermore, upon internal evaluation of the reviews provided, we kindly request you to disregard the reviewer report provided by Reviewer 2. No amendments are required in response to reviewer 2’s comments.

We look forward to receiving your revised manuscript.

Kind regards,

Avanti Dey, PhD

Staff Editor

PLOS Mental Health

Journal Requirements:

Additional Editor Comments (if provided):

Reviewers' comments:

Reviewer's Responses to Questions

**Comments to the Author**

Reviewer #2: All comments have been addressed

Reviewer #5: All comments have been addressed

publication criteria?

Reviewer #2: Yes

Reviewer #5: Yes

3. Has the statistical analysis been performed appropriately and rigorously?

Reviewer #2: Yes

Reviewer #5: N/A

4. Have the authors made all data underlying the findings in their manuscript fully available (please refer to the Data Availability Statement at the start of the manuscript PDF file)?

Reviewer #2: Yes

Reviewer #5: Yes

5. Is the manuscript presented in an intelligible fashion and written in standard English?

Reviewer #2: Yes

Reviewer #5: Yes

Reviewer #2: The Psychosocial Impact of Exercise as an Intervention for Persons Living With Obesity and Female Infertility: A Rapid Scoping Review and Pilot Study, is a commendable contribution to the growing body of literature on health and well-being. The authors have skillfully highlighted the intersection of physical and psychosocial health, offering valuable insights into how exercise interventions can support individuals dealing with obesity and infertility.

The combination of a scoping review and a pilot study is particularly noteworthy, as it provides both a broad synthesis of existing research and preliminary data that pave the way for future studies. The clarity of the methodology enhances the reliability of the findings, and the discussion section effectively connects these findings to practical applications.

Moreover, the article does an excellent job of addressing an often-overlooked area, the emotional and psychological challenges faced by individuals in these groups. The focus on evidence-based interventions like exercise is both timely and impactful. This study not only informs healthcare practitioners but also empowers patients by offering tangible strategies for improving quality of life.

Overall, the article stands out for its depth, rigor, and relevance, and it sets a strong foundation for further research in this critical area.

Reviewer #5: Although I am new to this manuscript, I can see that the authors have adequately addressed the previous concerns of Reviewers 1 to 4.

The paper is now solid. I am aware that the more people read a manuscript, the more different concerns appear. I still think some minor changes would make the manuscript even better. Here are my suggestions:

Abstract

Comment 1: Lines 38-39: “global stress” might not be a proper term here, since the FPI measures fertility-related stress. (The same is true for Lines 456-465, 600-624 etc., where the same term is used. I think the authors may mean “total/overall fertility stress”, based on summed FPI results.)

Background

Comment 2: Line 65: The “severity of infertility” – at first read this seems to be referring to the medical aspects of infertility. I suggest “the severe psychological impact of infertility”.

Comment 3: Lines 81-84: “discussion groups with patient partners on our research team” – This sentence is strange at this point of the paper (Introduction), where the reader is not supposed to know anything about the design (having discussion groups with patient partners). It strikes one as a quick and unexpected leap to methods and results. I suggest omitting this sentence altogether.

Comment 4: Lines 89-105: The authors succeeded in giving the rationale for the methodology used. This, however resulted in considerable redundancy in this part of the text. I suggest moving Lines 98-103 upwards (positioned after reference No 19), and shorten the yellow part (present Lines 89-98) to avoid repetitions. Also, a hypothesis for the pilot study should be clearly stated (as is present in a ‘covert form’ in Lines 533-535).

Comment 5: Line 91: Why was the BMI cut point of 28 kg/m2 used here? Confront also with Lines 188 and 194, where the cut point seems to be 30. Please consolidate and justify.

Methods

Comment 6: Lines 110-117: This is a very important part emphasizing the importance of listening to the unheard voice of a stigmatized group. However, I think it belongs to the Discussion. I think Lines 108-110 are enough at this point.

Comment 7: Line 110, and further along throughout the paper: The term ‘patient partners’ may be misleading, potentially also meaning “partners of patients” (e.g. in the framework of a heteroanamnestic discussion), feasible in the infertility context, in which “the patient” is usually the couple, and not an individual. I fully understand and welcome the positive message of patients being in partnership with researchers, however, I suggest another term (e.g. “partnering patient group”, “patient focus group”, “patient collaborators”) to avoid misunderstanding.

Comment 8: Lines 143-144: “Rapid scoping review results are summarized in table format in the results section of this paper.” I suggest omitting this sentence. It is obvious that results are presented in the Results section.

Comment 9: Table 1: In my understanding, inclusion criteria refer to the list of requirements that all study units (subjects or, as in this case, studies) have to meet in order to qualify for the study, and exclusion criteria to what units are then decided to be omitted from the initially included pool because their inclusion would distort the results. So, exclusion criteria are not the opposite, or lack of, inclusion criteria (as in Table 1), but a further refinement or narrowing of them, applied in a second step. Therefore, in my view, the second column of Table 1 is superfluous and wrongly called exclusion. In this study, an exclusion criterion would be, for example, exercise-based lifestyle interventions combined with psychological interventions (as opposed to those including only physical exercises), or participants with BMI values over 28 who have other chronic conditions, too, which may influence the effects of exercise, etc. I suggest that the authors rethink the presentation of the inclusion/exclusion criteria.

Good! See Lines 190-192, where the exclusion process is presented in the proper way (except for age)! These (i.e. unwillingness to delay fertility treatment, physical impairments limiting ability to participate, asthma or high blood pressure as relative criteria for exclusion) are the exclusion criteria to be presented in the right column, lines 1 and 4, of Table 1!

Comment 10: Line 150: Please use the full term of the acronym POR on first use.

Comment 11: Lines 164-169: The description of the questionnaires is incomplete (references, description of scales, sample items, validity and reliability data – like in Lines 393-397 regarding the HADS – are missing from here).

Comment 12: Line 207: “body weight” instead of “body weighted”.

Comment 13: Lines 209-210: I find it problematic that the exercise sessions were delivered by a kinesiologist and not a physiotherapist or a coach (trainer). This, of course, cannot be changed retrospectively, but I think this should be listed among the limitations.

Comment 14: Lines 220; 224-225: “The following questionnaires were administered: […] 7) Post-Intervention Semi-Structured Discussion Group” – the latter is not a questionnaire. Please reformulate.

Results

Comment 15: Lines 244-246: Unnecessary, can be deleted. Lines 246-250: information included in Figure 1, can be omitted, too. Figure 1: one number missing (n = ?).

Comment 16: Lines 266-315: I do not find it useful for the authors to give a one-by-one, detailed description of the studies reviewed. I think they should synthesize the findings, much like they do starting from Line 316. I suggest leaving out the individual descriptions and maybe make the summary in Lines 316-324 a bit longer and more comprehensive. (Lines 318-320: Unclear sentence.)

Comment 17: Lines 360-363: Can be omitted, has been already reported in Lines 164-169.

Comment 18: Table 4: The information on the questionnaires (such as in Lines 394-397 for the HADS; Lines 421-426 for the SF-12v2, etc.) should be included in the Methods section (Lines 164-169), not in the table legend or the Results. The Results section should only contain results.

Comment 19: Lines 393-491: I find the reporting on the results too lengthy and detailed. Since the tables contain all the relevant information on pre-post changes, the text should, again, be a concise synthesis of the results, e.g. “Physical component scores of health-related QoL increased in the majority of the patients (4 out of 7), with the rest stagnating (1/7) or decreasing (2/7)”.

Comment 20: Lines 499-500: Here, “instructors” is used in the plural, while earlier only a single kinesiologist was mentioned in the context of intervention delivery. Please be consistent.

Discussion

Comment 21: Lines 586-589: Mostly normal levels of hopelessness in the pilot study population may be attributed to the fact that the Beck Hopelessness Scale measures general and pervasive helplessness, while the patients in the discussion group may experience hopelessness focused on childbearing. This might be mentioned in the discussion.

Comment 22: Lines 610-611: “Unfortunately, no normative data for the FPI tool is available, limiting the ability to compare and interpret the present study findings” – Has the FPI not been validated in the mother tongue of the participants? Elsewhere in the paper it is stated that validated questionnaires were used (e.g. in Line 668). Pilot study results could be compared to validation results either in the authors’ own culture or, in lack of a validation process, in a related culture.

Comment 23: Line 648: I suggest “emerged” instead of “merged”.

Congratulations to the authors.

**Do you want your identity to be public for this peer review?** For information about this choice, including consent withdrawal, please see our Privacy Policy

Reviewer #2: **Yes: ** Jesan ara

Reviewer #5: **Yes: ** Judit Szigeti F.

---

## [Decision Letter · Decision Letter 2]

27 Jun 2025

PMEN-D-24-00233R2

The Psychosocial Impact of Exercise as an Intervention for Persons Living With Obesity and Female Infertility: A Rapid Scoping Review and Pilot Study

PLOS Mental Health

Dear Dr. Wadden,

Thank you for submitting your manuscript to PLOS Mental Health. After careful consideration, we feel that it has merit but does not fully meet PLOS Mental Health’s publication criteria as it currently stands. Therefore, we invite you to submit a revised version of the manuscript that addresses the points raised during the review process.

The manuscript has been evaluated by two reviewers, and their comments are available below.

Could you please ensure that you comment on the feedback provided by Reviewer 2, specifically on why the pre vs post p-values are mentioned but not reported?

Could you please carefully revise the manuscript to address all comments raised?

We look forward to receiving your revised manuscript.

Kind regards,

Johanna Pruller, Ph.D.

PLOS Staff Editor

PLOS Mental Health

Additional Editor Comments (if provided):

Reviewers' comments:

Reviewer's Responses to Questions

**Comments to the Author**

Reviewer #5: All comments have been addressed

Reviewer #6: (No Response)

publication criteria?

Reviewer #5: Yes

Reviewer #6: Partly

3. Has the statistical analysis been performed appropriately and rigorously?

Reviewer #5: N/A

Reviewer #6: Yes

4. Have the authors made all data underlying the findings in their manuscript fully available (please refer to the Data Availability Statement at the start of the manuscript PDF file)?

Reviewer #5: Yes

Reviewer #6: Yes

5. Is the manuscript presented in an intelligible fashion and written in standard English?

Reviewer #5: Yes

Reviewer #6: Yes

Reviewer #5: The authors properly addressed all of my comments and concerns. The paper's quality and clarity has indeed improved a lot. Congratulations!

Reviewer #6: Given the limited input, the systematic review was reasonable as per the PRISMA guidelines. Information was varied across the studies. Previous comments appear to have been addressed adequately. They used the dual methodological approach by combining a rapid scoping review and a pilot study to strengthen the validity of their findings while addressing unmet needs in the literature. The results of both methodologies is seen in the text and tables. Since this is a descriptive overview the ‘strength’ of the validity is not really quantitative. However, the interpretations allowed for an attempt to summarize the improvements on some domains and the ‘not’ improvements on others.

As per the authors, due to the small sample size (n=11), statistical analysis of pilot study data is limited to descriptive statistics. This is clear from the tables presented summarizing the endpoints of interest. Confidence intervals and P-values (and perhaps even the standard deviations in some cases) for the results are not reliable due to the small sample size of the pilot study. Many before and after statistics are presented, but this reviewer did not see many pre vs. post p-values ( even non parametrically due to the small sample size). Why mention them and not report them and then caution against their interpretation? Did this reviewer miss something? The authors did much work and the effort is evident from the material presented. The conclusions followed from the observational aspects of the limited data. They did well to note the limitations of the approach throughout.

**Do you want your identity to be public for this peer review?** For information about this choice, including consent withdrawal, please see our Privacy Policy

Reviewer #5: **Yes: ** Judit Szigeti F.

Reviewer #6: No

---

## [Editor Report · Decision Letter 3]

13 Aug 2025

PMEN-D-24-00233R3

The Psychosocial Impact of Exercise as an Intervention for Persons Living With Obesity and Female Infertility: A Rapid Scoping Review and Pilot Study

PLOS Mental Health

Dear Dr. Wadden,

Thank you for submitting your manuscript to PLOS Mental Health. After careful consideration, we feel that it has merit but does not fully meet PLOS Mental Health’s publication criteria as it currently stands. Therefore, we invite you to submit a revised version of the manuscript that addresses the points raised during the review process.

The reviewer has raised concerns regarding the reporting of before and after statistics. Could you please carefully respond to the comments provided? 

We look forward to receiving your revised manuscript.

Kind regards,

Johanna Pruller, Ph.D.

PLOS Staff Editor

PLOS Mental Health
---

## [Decision Letter · Decision Letter 4]

6 Oct 2025

The Psychosocial Impact of Exercise as an Intervention for Persons Living With Obesity and Female Infertility: A Rapid Scoping Review and Pilot Study

PMEN-D-24-00233R4

Dear Dr. Wadden,

We are pleased to inform you that your manuscript 'The Psychosocial Impact of Exercise as an Intervention for Persons Living With Obesity and Female Infertility: A Rapid Scoping Review and Pilot Study' has been provisionally accepted for publication in PLOS Mental Health.

Best regards,

Karli Montague-Cardoso

Staff Editor

PLOS Mental Health

Reviewer Comments (if any, and for reference):

Reviewer's Responses to Questions

**Comments to the Author**

Reviewer #6: All comments have been addressed

publication criteria?

Reviewer #6: (No Response)

3. Has the statistical analysis been performed appropriately and rigorously?

Reviewer #6: (No Response)

4. Have the authors made all data underlying the findings in their manuscript fully available (please refer to the Data Availability Statement at the start of the manuscript PDF file)?

Reviewer #6: (No Response)

5. Is the manuscript presented in an intelligible fashion and written in standard English?

Reviewer #6: (No Response)

Reviewer #6: The authors have made a good case that the limitations noted, especially the unexpected anxiety, have allowed for feasibility and what to incorporate into a future study. They have stated honestly in the statistical analysis section that these results should not be interpreted as statistically significant findings, but rather as preliminary observations appropriate for a pilot study.

**Do you want your identity to be public for this peer review?** For information about this choice, including consent withdrawal, please see our Privacy Policy

Reviewer #6: No
